# Robust Rent Division

**Dominik Peters**
CNRS, Université Paris Dauphine–PSL
dominik@lamsade.fr

**Ariel D. Procaccia**
Harvard University
arielpro@seas.harvard.edu

**David Zhu**
Harvard University
david.zhu@gmail.com

## Abstract

In fair rent division, the problem is to assign rooms to roommates and fairly split the rent based on roommates' reported valuations for the rooms. Envy-free rent division is the most popular application on the fair division website Spliddit. The standard model assumes that agents can correctly report their valuations for each room. In practice, agents may be unsure about their valuations, for example because they have had only limited time to inspect the rooms. Our goal is to find a robust rent division that remains fair even if agent valuations are slightly different from the reported ones. We introduce the lexislack solution, which selects a rent division that remains envy-free for valuations within as large a radius as possible of the reported valuations. We also consider robustness notions for valuations that come from a probability distribution, and use results from learning theory to show how we can find rent divisions that (almost) maximize the probability of being envy-free, or that minimize the expected envy. We show that an almost optimal allocation can be identified based on polynomially many samples from the valuation distribution. Finding the best allocation given these samples is NP-hard, but in practice such an allocation can be found using integer linear programming.

## 1 Introduction

The literature on fair division of resources has produced allocation mechanisms for many domains, such as course allocation, indivisible goods, chores, house assignment, and the selection of citizens' assemblies [Budish, 2011, Caragiannis et al., 2019, Moulin, 2019, Flanigan et al., 2021]. But arguably the most widely *used* example is *rent division*: this is the most popular application on the fair division website spliddit.org [Goldman and Procaccia, 2014], where it has been used more than 30,000 times since its launch in 2014.

Rent division deals with the common situation where a group of $n$ future roommates are planning to move into a house or apartment which has $n$ rooms, one for each roommate. They will split the rent payments among themselves. The roommates may differ in how much they are willing to pay for different rooms. Given the room valuations of each roommate, our task is to assign the rooms, and to decide how to split the rent. We wish to do this fairly, and so we will choose an allocation that is *envy-free*: no roommate would strictly prefer to get another room, given the prices we have assigned to those rooms. Such an allocation is guaranteed to exist [Svensson, 1983].

Let us consider an example with $n = 3$ roommates, and let the total rent be \$1000. Table 1 shows the valuation that each agent assigns to each room. Given this information, the algorithm in use on Spliddit will assign Room 1 to Alice, Room 2 to Bob, and Room 3 to Charlie, charging them \$100, \$500, and \$400 respectively. This allocation is envy-free under the assumption (which we will make

36th Conference on Neural Information Processing Systems (NeurIPS 2022).

|         | Room 1 | Room 2 | Room 3 |
|---------|--------|--------|--------|
| Alice   | **300** | 400   | 300    |
| Bob     | 300    | **700** | 0      |
| Charlie | 300    | 100    | **600** |
| Spliddit | 100   | 500    | 400    |
| Lexislack | 200  | 450    | 350    |

Table 1: Example of valuations. In any envy-free allocation, Alice gets room 1, Bob gets room 2, and Charlie gets room 3. The lower rows display the price vectors selected by Spliddit's rule (maximin) and by our lexislack rule.

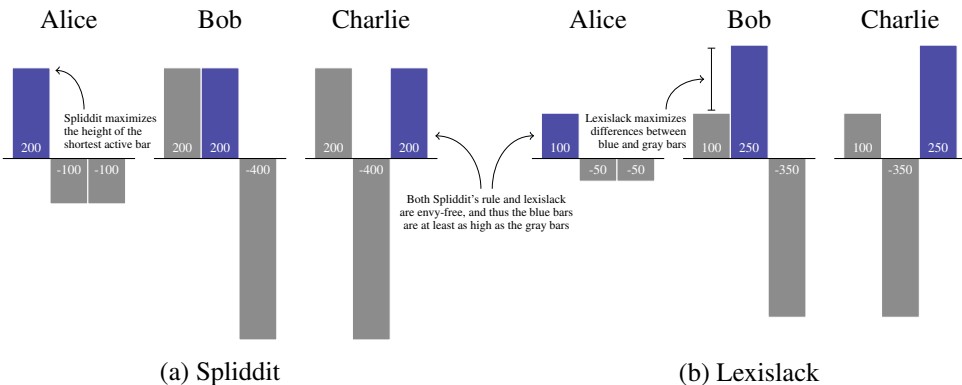

(a) Spliddit                    (b) Lexislack

Figure 1: Illustration of the example in Table 1. For Spliddit's rule and for our lexislack rule, we show the quasilinear utility (value minus price) that each agent gets from each room. The bar corresponding to the room assigned to the agent is shaded in blue. Both rules are envy-free, and thus the blue bars are at least high as the gray bars. Subject to envy-freeness, Spliddit's rule maximizes the height of the shortest blue bar. Lexislack maximizes the differences in height between the blue and the gray bars.

throughout) that agents have *quasilinear utilities*: their utility under an allocation is the value of their room minus its price. For example, Alice has utility $300 - 100 = 200$. She does not envy the others: Bob's room would give her only utility $400 - 500 < 200$, and Charlie's room $300 - 400 < 200$.

On a typical instance, there are infinitely many allocations that are envy-free. Spliddit's algorithm chooses the one that maximizes the utility of the worst-off agent, subject to envy-freeness [Gal et al., 2017]. This is known as the *maximin* rule. In optimizing this objective, Spliddit might choose an outcome that is only barely envy-free. In the example, Bob has utility $700 - 500 = 200$, but he would gain the same utility from having Alice's room: $300 - 100$. If, upon moving in, Bob discovers a defect in his room and now only values it at 600, say, then he would envy Alice. Thus, the envy-freeness of Spliddit's allocation is not robust.

We study the rent division problem with the goal of finding allocations that are robustly envy-free, in the sense that they remain envy-free even if valuations change slightly. For this, we introduce the *lexislack rule*, which selects an envy-free allocation where the minimum "slack" (the amount by which agent $i$ prefers her allocation to agent $j$'s) is maximized lexicographically. This produces an allocation that remains envy-free for all valuation profiles that are within a maximally large $\ell_1$-radius of the reported profile. In the example of Table 1, the lexislack rule assigns the rooms in the same way as does Spliddit, but charges the roommates \$200, \$450, and \$350. With these prices, each agent prefers their allocation to any other agent's by at least 150 (see Figure 1). This means that even after Bob's adjustment to 600, he does not envy Alice. We show that the lexislack rule always selects an essentially unique outcome, which can be found in polynomial time by linear programming.

This notion of robustness may not always be appropriate. Consider two perturbations with equal $\ell_1$-distance to the reported valuations: one changes agent $i$'s valuations for all rooms by a small amount, the other changes $i$'s valuation for one room by a large amount. Lexislack places equal importance on them. But the former perturbation seems more likely: even if a player is uncertain about the value of a room, that value is more likely to be close to their best estimate than further

away. Thus, arguably, different valuation profiles should be weighted differently: we do not want to sacrifice envy-freeness for a likely perturbation in order to obtain it for an unlikely perturbation.

To capture this idea, we propose to add noise — such as Gaussian noise — around the reported valuations. This way, we impute a probability distribution $\mathcal{D}$ over valuations. In this setting, our interpretation of robustness is to look for allocations that are envy-free with *maximum probability*. However, it is not clear how one could efficiently find the most robust allocation given the noisy valuations. As part of our methodological contribution, we propose an approach based on synthetic sampling. Specifically, we sample a number of valuation profiles from $\mathcal{D}$, and then find an allocation that is optimal on this sample using integer linear programming (ILP). By calculating the VC dimension of the space of rent divisions, we give polynomial sample complexity bounds that show how many samples are sufficient so that this approach identifies an almost optimal allocation with high probability. Note that the samples are synthetic, but low sample complexity is crucial nevertheless: a small number of samples leads to a sufficiently small ILP that, as we show, can be optimally solved in practice (even though we prove that the problem is NP-complete).

We also show that one can use the sampling approach to find an allocation that minimizes the expected amount by which one agent envies another. In contrast to maximizing the probability of envy-freeness, the minimum envy objective places more emphasis on avoiding bad violations of envy-freeness.

An advantage of our sampling-based approach is that it is very general and does not place any restrictions on the distribution $\mathcal{D}$. Our algorithms could also be used for rent division problems with uncertainty, where agents might explicitly report distributions over their valuations. For example, a Spliddit-like user interface could let agents report their valuations as a *range* rather than a number.

We end with some experiments on data taken from Spliddit. They suggest that our three new rules significantly outperform the Spliddit maximin rule on robustness metrics. Interestingly, the lexislack solution does comparably well to the rules based on sampling. Given its conceptual simplicity and easy computation, this suggests lexislack as a good rule when robustness is desired.[1]

**Related Work**

The rent division model is well-studied in the economics literature [Svensson, 1983, Alkan et al., 1991, Aragones, 1995, Su, 1999, Velez, 2018], often without assuming quasilinear utilities. That literature includes results on the structure of the envy-free set and about strategic aspects. Computer scientists have studied the computation of allocation rules [Gal et al., 2017, Procaccia et al., 2018]. Bei et al. [2021] study a generalization of the rent division problem.

Robustness has been studied in several areas of computational social choice, such as in voting [Shiryaev et al., 2013], in committee elections [Bredereck et al., 2021, Gawron and Faliszewski, 2019, Misra and Sonar, 2019], and in stable matching [Chen et al., 2019, Mai and Vazirani, 2018]. We are not aware of such work for fair division, though Menon and Larson [2020] study a related problem of "stability" which requires that the allocation should not change much if valuations change slightly. For rent division, a blog post by Critch [2015] argues in favor of aiming for robustness in the rent division problem. Critch [2015] implemented an algorithm for robust rent division that appears in experiments to maximize the slack, but it differs from the lexislack rule, and no theoretical analysis of this algorithm is available.

Our sampling-based approach is conceptually related to work on data-driven algorithm design [Balcan, 2020], which typically seeks to optimize the hyperparameters of an algorithm with respect to an underlying distribution over instances, based on samples. One thing that distinguishes our distributional setting is that we are using the samples to optimize a single solution to our problem. Computational hardness results for problems similar to our sample-based optimization problems have been obtained for stable matching and for Pareto-optimal assignment [Aziz et al., 2019, 2020].

## 2 Preliminaries: Rent Division

Let $n \in \mathbb{N}$ and write $[n] = \{1, \ldots, n\}$. Let $N = [n]$ be a set of $n$ agents, and let $R = [n]$ be a set of $n$ rooms. Without loss of generality, we let the total rent be 1. A *(valuation) profile* $v = (v_{ir})_{i \in N, r \in R}$ is a collection of values $v_{ir} \in \mathbb{Q}_+$, one for each agent $i \in N$ and each room $r \in R$.

---

[1]A simple online tool to compare the lexislack and maximin rules is available at https://pref.tools/rent.

A *room assignment* is a bijection $\sigma : N \to R$, so that agent $i$ is assigned room $\sigma(i)$. Given a valuation profile $v$, we say that $\sigma$ is *optimal* if it maximizes utilitarian social welfare $\sum_{i \in N} v_{i\sigma(i)}$. An *allocation* $(\sigma, p)$ is a room assignment $\sigma$ together with a payment vector $p = (p_1, \ldots, p_n) \in \mathbb{R}^n$ with $\sum_{r \in R} p_r = 1$, where $p_r$ is the rent of room $r$. (The value $p_r$ is usually non-negative.)

We assume that agents have *quasilinear utilities*. This means that if $v$ is a valuation profile and $(\sigma, p)$ is an allocation, then agent $i$'s utility in this allocation is $v_{i\sigma(i)} - p_{\sigma(i)}$, i.e., the valuation of $i$ for her room $\sigma(i)$ minus the room's rent. An allocation $(\sigma, p)$ is *envy-free* if $v_{i\sigma(i)} - p_{\sigma(i)} \geqslant v_{ir} - p_r$ for all $i \in N$ and $r \in R$, so that each agent $i$ weakly prefers her allocation to receiving any other room.

A *solution* is a function that given a valuation profile, selects a set of allocations (usually a singleton, but ties may occur). A solution is *essentially single-valued* if when it selects more than one allocation, then all agents are indifferent between them: every agent gets the same utility from all tied allocations.

The following facts are well-known [see, e.g., Velez, 2018]. We include proofs in Appendix D.1.

**Theorem 2.1.** *(a) For every optimal room assignment $\sigma$, there are prices $p$ so that $(\sigma, p)$ is envy-free.*
*(b) If $(\sigma, p)$ is envy-free then $\sigma$ is optimal.*
*(c) Let $\sigma_1$, $\sigma_2$ be optimal room assignments, and let $(\sigma_1, p)$ be an envy-free allocation. Then $(\sigma_2, p)$ is also an envy-free allocation, with all agents indifferent between the two: $v_{i\sigma_1(i)} - p_{\sigma_1(i)} = v_{i\sigma_2(i)} - p_{\sigma_2(i)}$ for all $i \in N$.*

Theorem 2.1(a) implies that an envy-free allocation exists for all valuation profiles. We can compute one in polynomial time: find an optimal room assignment $\sigma$ using bipartite matching, then use linear programming to find prices $p$ that make the allocation envy-free [Gal et al., 2017]. Theorem 2.1(c) implies that when selecting among envy-free allocations, we can restrict attention to any fixed $\sigma$ and only vary the price vector $p$. By Theorem 2.1(c), all utility vectors achievable in an envy-free allocation are achieved by allocations of this form.

## 3 The Lexislack Solution

We start by considering a common form of robustness: we look for allocations that remain fair for all valuations that are within some radius of input valuations, for as large a radius as possible. Thus, unlike in later sections, we do not assume that valuations come from a probability distribution.

Let $v$ be a valuation profile, fixed throughout. Let $(\sigma, p)$ be an allocation. For $i \in N$ and $r \in R$, let

$$\Delta_{ir}(\sigma, p) = (v_{i\sigma(i)} - p_{\sigma(i)}) - (v_{ir} - p_r).$$

Then define the *slack* of this allocation as

$$\mathsf{slack}(\sigma, p) = \min_{i \in N} \min_{r \neq \sigma(i)} \Delta_{ir}(\sigma, p).$$

Thus, an allocation has positive slack if every agent strictly prefers their allocation to all other agents' allocations. An allocation $(\sigma, p)$ is envy-free if and only if $\mathsf{slack}(\sigma, p) \geqslant 0$.

Slack is a measure of how robustly fair an allocation is, which we formalize in the following result.

**Proposition 3.1.** *Let $(\sigma, p)$ be an envy-free allocation with $\mathsf{slack}(\sigma, p) = s \geqslant 0$. If $v'$ is a valuation profile that is $s$-close to $v$ in the sense that*

$$\|v_i - v_i'\|_1 = \sum_{r \in R} |v_{ir} - v_{ir}'| \leqslant s$$

*for all $i \in N$, then $(\sigma, p)$ is also envy-free under $v'$.*

*Proof.* Let $i, j \in N$. Then $\sum_{r \in R} |v_{ir} - v_{ir}'| \leqslant s$ implies

$$(v_{i\sigma(i)} - v_{i\sigma(i)}') + (v_{i\sigma(j)}' - v_{i\sigma(j)}) \leqslant s \tag{1}$$

Adding $p_{\sigma(j)} - p_{\sigma(i)}$ to both sides and rearranging, we get

$$(v_{i\sigma(i)}' - p_{\sigma(i)}) - (v_{i\sigma(j)}' - p_{\sigma(j)}) \geqslant (v_{i\sigma(i)} - p_{\sigma(i)}) - (v_{i\sigma(j)} - p_{\sigma(j)}) - s \geqslant 0$$

where the last inequality is by definition of slack. Thus, $i$ does not envy $j$ under $v'$. Since $i$ and $j$ were arbitrary, it follows that $(\sigma, p)$ is envy-free under $v'$. $\square$

One can also prove variants of Proposition 3.1. For example, $\|v_i - v_i'\|_\infty \leqslant s/2$ also implies (1).[2]

If we wish to ensure robustness in a sense like in Proposition 3.1, this suggests the following rule:

$$\text{maxislack}(v) = \text{argmax}_{(\sigma,p)} \text{slack}(\sigma, p).$$

This rule always selects an envy-free allocation: since envy-free allocations exist for every $v$, there exists an allocation with non-negative slack, and hence the maxislack solution also has non-negative slack. A maxislack solution can be found in polynomial time by computing an optimal assignment $\sigma$ and then solving the following LP:

$$\max L \text{ subject to } (v_{i\sigma(i)} - p_{\sigma(i)}) - (v_{i\sigma(j)} - p_{\sigma(j)}) \geqslant L \;\forall i \neq j \text{ and } \sum_{r \in R} p_r = 1.$$

However, there are a few drawbacks to the maxislack rule. First, the rule is not essentially single-valued: there may be several maxislack allocations which induce different utilities. This is unlike Spliddit's maximin rule which is essentially single-valued [Alkan et al., 1991]. Second, there may be maxislack allocations that do not maximize robustness for *all* agents. To see this, suppose that two agents $i_1$ and $i_2$ agree on the valuation of every room. Then in any envy-free allocation, the utility they assign to the two bundles allocated to them is equal. Hence the maximum slack attainable is 0, and so *every* envy-free allocation is maxislack. However, there may be allocations for which the slack between other pairs of agents is larger than 0, and such allocations are more robustly fair.

In this spirit, to obtain robustness for a larger collection of agents (or of agent pairs), we can refine the maxislack solution using a leximin strategy. We call the resulting solution the *lexislack rule*. The lexislack rule selects an allocation $(\sigma, p)$ that maximizes the smallest of the $n^2$ values $(\Delta_{ir}(\sigma, p))_{i \in N, r \in R}$, and subject to that maximizes the second-smallest of these values, and so on.

In contrast to maxislack, the lexislack rule is essentially single-valued. The proof is in Appendix D.2.

**Theorem 3.2.** *The lexislack rule is essentially single-valued.*

In addition, this rule remains efficiently computable.

**Theorem 3.3.** *A lexislack allocation can be found in poly time by solving $O(n^4)$ linear programs.*

*Proof sketch.* This can be done using standard techniques [see Kurokawa et al., 2018, Section 5]. We give an overview of the algorithm. Start by computing an optimal $\sigma$. We will decide on the best value of $\Delta_{ir}$ one-by-one. Let $F \leftarrow \emptyset$ be the set of $(i, r)$ pairs for which we have fixed their value. Use linear programming to find a price vector such that $(\sigma, p)$ maximizes the smallest of the non-fixed values $\Delta_{ir}$, subject to keeping the other $\Delta_{ir}$ at their fixed value. Say the optimum is $L$. Now we need to find a pair $(i, r) \notin F$ such that necessarily $\Delta_{ir} = L$ in any lexislack allocation. This can again be done by linear programs that check whether it is possible that $\Delta_{ir} > L$. One can show that at least one such pair $(i, r) \notin F$ must exist; we then add it to $F$ and fix its value to $L$, and repeat. $\qquad\square$

## 4 Maximizing Probability of Envy-Freeness

In the previous section, we defined robustness using a measure of closeness based on the $\ell_1$-distance. We now look at a more flexible model where true valuations are assumed to be noisy perturbations of the reported ones. A distribution $\mathcal{D}$ over valuations $v$, therefore, is obtained by asking agents for valuations, and then adding noise (e.g., Gaussian or uniform) around those valuations. Our goal will be to find an allocation $(\sigma, p)$ that maximizes the probability of being envy-free with respect to $\mathcal{D}$, i.e., one that maximizes

$$\text{EFrate}_{\mathcal{D}}(\sigma, p) = \text{Pr}_{v \sim \mathcal{D}}[(\sigma, p) \text{ is envy-free under } v].$$

Our algorithmic approach for finding an allocation with high probability of envy-freeness is to obtain a sample $S$ of $m$ valuation profiles sampled from $\mathcal{D}$, and to compute an allocation that is envy-free on the most profiles in $S$, i.e., one that maximizes

$$\text{EFrate}_S(\sigma, p) = \tfrac{1}{m} \cdot |\{v \in S : (\sigma, p) \text{ is envy-free under } v\}|.$$

If the number $m$ of samples is sufficiently high, we may hope that the best allocation on the sample $S$ is also approximately the best on the distribution $\mathcal{D}$. In this section, we will give a bound for the sample size $m$ to be sufficient to ensure this property, and then we will discuss the computational problem of finding the best allocation for a given sample.

---

[2]In future work, it may be interesting to study rules that explicitly maximize robustness defined with respect to $\ell_\infty$-distance rather than $\ell_1$.

## 4.1 Sample Complexity

In this section, we will give an upper bound on the number of samples required to guarantee that the allocation that maximizes $\mathsf{EFrate}_S$ also (almost) maximizes $\mathsf{EFrate}_\mathcal{D}$, with high probability.

**Theorem 4.1.** *Let $\varepsilon, \delta > 0$. There is a value $m \in \mathbb{N}$ with*

$$m = O\left(\frac{n^2 \log n + \log(1/\delta)}{\varepsilon^2}\right)$$

*such that for every probability distribution $\mathcal{D}$ over valuation profiles, if $S$ is a collection of at least $m$ samples drawn i.i.d. from $\mathcal{D}$, and $(\sigma^*, p^*)$ is the allocation that maximizes $\mathsf{EFrate}_S$, then with probability at least $1 - \delta$,*

$$\mathsf{EFrate}_\mathcal{D}(\sigma^*, p^*) \geqslant \max_{(\sigma, p)} \mathsf{EFrate}_\mathcal{D}(\sigma, p) - \varepsilon.$$

We prove this theorem by adapting standard tools from learning theory. Let $X$ be any set, with an unknown ground truth labeling $\tau : X \to \{0, 1\}$. A *hypothesis* is a function $h : X \to \{0, 1\}$. Given a *sample* $S = (x_1, \ldots, x_m)$ of $m$ elements of $X$ (not necessarily distinct), write $\mathsf{err}_S(h) = \frac{1}{m}|\{x_i : h(x_i) \neq \tau(x_i)\}|$ for the fraction of samples that $h$ labeled incorrectly. For a probability distribution $\mathcal{D}$ over $X$, write $\mathsf{err}_\mathcal{D}(h) = \Pr_{x \sim D}[h(x) \neq \tau(x)]$ for the probability that $h$ incorrectly labels a point sampled from $\mathcal{D}$.

A *hypothesis class* $\mathcal{H}$ is a set of hypotheses. Given a random sample $S$ drawn i.i.d. from $\mathcal{D}$, and knowledge of the true labeling $\tau$ of those samples, our goal is to find a hypothesis $h \in H$ that approximately minimizes $\mathsf{err}_\mathcal{D}(h)$, with high probability. Note that the ground truth $\tau$, interpreted as a hypothesis, need not be a member of $\mathcal{H}$. In learning theory, this setup corresponds to "agnostic PAC learning", where the "realizability assumption" is not required to hold [Shalev-Shwartz and Ben-David, 2014, Section 3.2].

We say that a set $C \subseteq X$ is *shattered* by $\mathcal{H}$ if for all $S \subseteq C$, there exists $h \in \mathcal{H}$ with $h(x) = 1$ if $x \in S$ and $h(x) = 0$ if $x \in C \setminus S$. In other words, if we restrict the hypotheses in $\mathcal{H}$ to the set $C$, then all possible labelings of $C$ are part of $\mathcal{H}$. The *VC dimension* $\mathsf{VCdim}(\mathcal{H})$ of $\mathcal{H}$ is the cardinality of the largest subset of $X$ that is shattered by $\mathcal{H}$. We are interested in VC dimension due to the following standard result, adapted from Shalev-Shwartz and Ben-David [2014, Theorem 6.8], which says that PAC learning is possible on hypothesis classes of finite VC dimension.

**Theorem 4.2.** *Let $\varepsilon, \delta > 0$. Let $\mathcal{H}$ be a hypothesis class with $\mathsf{VCdim}(\mathcal{H}) = d$. Then there exists a value $m \in \mathbb{N}$ with*

$$m = O\left(\frac{d + \log(1/\delta)}{\varepsilon^2}\right)$$

*such that for every probability distribution $\mathcal{D}$ over $X$, if $S$ is a collection of at least $m$ samples drawn i.i.d. from $\mathcal{D}$, and $h^* \in H$ is the hypothesis that minimizes $\mathsf{err}_S$, then with probability at least $1 - \delta$,*

$$\mathsf{err}_\mathcal{D}(h^*) \leqslant \min_{h \in \mathcal{H}} \mathsf{err}_\mathcal{D}(h) + \varepsilon.$$

For our application, we let $X$ be the set of all valuation profiles $v$. The "correct" labeling is $\tau(v) = 1$ for all $v$. We identify allocations with hypotheses: For an allocation $(\sigma, p)$, we define the hypothesis $h_{(\sigma, p)}$ so that for each $v$,

$$h_{(\sigma, p)}(v) = \begin{cases} 1 & \text{if } (\sigma, p) \text{ is envy-free under } v, \\ 0 & \text{otherwise.} \end{cases}$$

By these definitions, we have that for all $S$ and $\mathcal{D}$,

$$\mathsf{EFrate}_S(\sigma, p) = 1 - \mathsf{err}_S(h_{(\sigma, p)}), \text{ and}$$
$$\mathsf{EFrate}_\mathcal{D}(\sigma, p) = 1 - \mathsf{err}_\mathcal{D}(h_{(\sigma, p)}).$$

We study the hypothesis class $\mathcal{H}$ of all such hypotheses:

$$\mathcal{H} = \{h_{(\sigma, p)} : \text{allocations } (\sigma, p)\}.$$

To bound its VC dimension, the following result is useful:

**Lemma 4.3** (Shalev-Shwartz and Ben-David, 2014, Exercise 6.11). *Let $\mathcal{H}_1, \ldots, \mathcal{H}_t$ be hypothesis classes over $X$, with $\mathsf{VCdim}(\mathcal{H}_i) \leqslant d$ for each $i = 1, \ldots, t$. Then*

$$\mathsf{VCdim}(\mathcal{H}_1 \cup \cdots \cup \mathcal{H}_t) \leqslant 4d \log(2d) + 2\log(t).$$

We can now bound the VC dimension of $\mathcal{H}$.

**Lemma 4.4.** $\mathsf{VCdim}(\mathcal{H}) = O(n^2 \log n)$.

*Proof.* For each room assignment $\sigma$, define the hypothesis class $\mathcal{H}_\sigma = \{h_{(\sigma,p)} : p \in \mathbb{R}^n\}$ corresponding to allocations whose room assignment is $\sigma$. Then $\mathcal{H} = \bigcup_\sigma \mathcal{H}_\sigma$ where the union ranges over all room assignments. We will show that $\mathsf{VCdim}(\mathcal{H}_\sigma) \leqslant n^2$ for each $\sigma$. Since there are $n!$ different room assignments and $\log n! = O(n \log n)$, it follows from Lemma 4.3 that $\mathsf{VCdim}(\mathcal{H}) = O(n^2 \log n)$, as required.

Let $\sigma$ be a room assignment. Without loss of generality assume that $\sigma(i) = i$. Let $d \geqslant n^2 + 1$. Consider a collection of $d$ distinct valuation profiles $v^{(1)}, \ldots, v^{(d)}$. We show that this collection cannot be shattered by $\mathcal{H}_\sigma$.

For $i, j \in N$, say $v^{(k)}$ is *uniquely restricting for* $(i, j)$ if

$$v_{ij}^{(k)} - v_{ii}^{(k)} > v_{ij}^{(\ell)} - v_{ii}^{(\ell)} \quad \text{for all } \ell \neq k.$$

Thus, such a profile uniquely maximizes the amount by which agent $i$ prefers $j$'s room to her own room, ignoring prices. Clearly, for any pair $i, j \in N$, at most one profile can be uniquely restricting for it. Since there are $n^2$ many pairs $(i, j)$ and $d > n^2$, there is at least one profile which is not uniquely restricting for any pair, say $v^{(1)}$.

We now ask if there is an allocation $(\sigma, p)$ that is envy-free under $v^{(2)}, \ldots, v^{(d)}$, but not envy-free under $v^{(1)}$. We show that the answer is no, so $\mathcal{H}_\sigma$ fails to shatter this collection.

Assume for a contradiction that $(\sigma, p)$ is such an allocation. Since it is not envy-free under $v^{(1)}$, there is a pair $i, j \in N$ with $v_{ij}^{(1)} - p_j > v_{ii}^{(1)} - p_i$ or equivalently

$$v_{ij}^{(1)} - v_{ii}^{(1)} > p_j - p_i. \tag{2}$$

As $v^{(1)}$ is not uniquely restricting for $(i, j)$, for some $\ell \neq 1$,

$$v_{ij}^{(\ell)} - v_{ii}^{(\ell)} \geqslant v_{ij}^{(1)} - v_{ii}^{(1)}. \tag{3}$$

Combining (2) and (3), it follows that $v_{ij}^{(\ell)} - v_{ii}^{(\ell)} > p_j - p_i$. Thus, $(\sigma, p)$ is not envy-free under $v^{(\ell)}$, a contradiction. $\square$

Our main result in this section, Theorem 4.1, now follows immediately from Theorem 4.2.

## 4.2 Computational Complexity

To make use of Theorem 4.1, we need an algorithm that, given a collection $S = (v^{(1)}, \ldots, v^{(m)})$ of valuation profiles sampled from $\mathcal{D}$, finds an allocation that maximizes $\mathsf{EFrate}_S(\sigma, p)$. This problem can be encoded as an integer linear program via standard encoding techniques, using binary variables $x_{ir}$ encoding that agent $i$ receives room $r$, continuous variables $p_r$ encoding the prices, and a binary variable $y_\ell$ for each sample $\ell \in [m]$, indicating whether the produced allocation will be envy-free under $v^{(\ell)}$. The full encoding appears in Appendix B.

Instead of an ILP approach, can we hope for a polynomial time algorithm finding the best allocation? Let us formulate our optimization problem as a decision problem as follows.

---

EF-RATE MAXIMIZATION
**Input:** Set $N$ of agents, set $R$ of rooms, a list of $m$ valuation profiles $v^{(1)}, \ldots, v^{(m)}$, number $B$.
**Question**: Does there exist an allocation that is envy-free for at least $B$ of the $m$ valuation profiles?

---

Unfortunately, this problem is computationally hard. We prove this by a reduction from CLIQUE, given in Appendix D.3.

**Theorem 4.5.** EF-RATE MAXIMIZATION *is NP-complete.*

There are two sources of computational difficulty for solving EF-RATE MAXIMIZATION: we have to decide on one of the $n!$ possible room assignments, and we have to decide on which subset of valuation profiles we are aiming to be envy-free on. But in practice, there is a way to avoid the first source of hardness. Suppose the $m$ valuation profiles are sampled from a *continuous* distribution $\mathcal{D}$. Then with probability 1, for each sampled profile $v^{(\ell)}$ there is a *unique* optimal room assignment $\sigma^{(\ell)}$. Any solution to the EF-rate maximization problem must use a room assignment that is optimal for at least one of the given valuation profiles. Thus, at most $m$ different room assignments are candidates, and we can find an optimal solution using $m$ calls to the following problem (one call for each candidate assignment $\sigma^{(\ell)}$):

---

EF-RATE MAXIMIZATION WITH FIXED ASSIGNMENT
**Input:** A list of $m$ valuation profiles $v^{(1)}, \ldots, v^{(m)}$, number $B$, room assignment $\sigma$.
**Question**: Is there a price vector $p$ such that $(\sigma, p)$ is envy-free for at least $B$ of the $m$ valuation profiles?

---

Unfortunately, this version of the problem is also hard, and so this trick for continuous distributions does not help. We prove this by reduction from the feedback arc set problem in Appendix D.4.

**Theorem 4.6.** EF-RATE MAXIMIZATION WITH FIXED ASSIGNMENT *is NP-complete.*

Nevertheless, as we show in Section 6, we can solve this problem in practice using integer linear programming (ILP). The reason this is possible is that the sample complexity is relatively low, leading to an ILP of practical size.

## 5 Minimizing Expected Envy

In Section 4, we defined robust envy-freeness as allocations that have a high probability of being envy-free when valuations come from a given distribution $\mathcal{D}$. In this section, we consider a different objective function that is more fine-grained. In measuring the probability of envy-freeness, we implicitly treat all failures of envy-freeness equally. We will now minimize *expected envy*, which treats cases where one agent envies another by a lot as more severe.

Given a valuation profile $v$ and an allocation $(\sigma, p)$, we define the allocation's *(maximum) envy*, $\mathsf{envy}_v(\sigma, p)$, to be

$$\max\left\{0, \max_{i,j \in N} \left[(v_{i\sigma(j)} - p_{\sigma(j)}) - (v_{i\sigma(i)} - p_{\sigma(i)})\right]\right\}.$$

This quantity, which is related to slack as considered in Section 3, measures the biggest amount by which one agent prefers another's bundle. In principle one could allow negative values of $\mathsf{envy}_v(\sigma, p)$ for allocations that have positive slack, but we chose to force these values to be non-negative, since our focus is on avoiding envy. Note that an allocation is envy-free if and only if $\mathsf{envy}_v(\sigma, p) = 0$.

Our goal in this section is to find an allocation minimizing the expected envy with respect to $\mathcal{D}$, defined as

$$\mathsf{envy}_{\mathcal{D}}(\sigma, p) = \mathbb{E}_{v \sim \mathcal{D}}[\mathsf{envy}_v(\sigma, p)].$$

Our approach will be similar to before: we obtain a sufficiently large sample $S$ of $m$ profiles from $\mathcal{D}$ and select the allocation that does best on the sample, i.e. it minimizes

$$\mathsf{envy}_S(\sigma, p) = \tfrac{1}{m} \sum_{v \in S} \mathsf{envy}_v(\sigma, p).$$

### 5.1 Sample Complexity

For stating our sample complexity bound, we assume that valuations $v$ are normalized: let $v_{ir} \geqslant 0$ for all $i \in N$ and $r \in R$, and $\sum_{r \in R} v_{ir} = 1$ for all $i \in N$. We are going to prove the following result:

**Theorem 5.1.** *Let $\varepsilon, \delta > 0$, and let $\mathcal{D}$ be a distribution. If we draw $m = O(\frac{n}{\varepsilon^2} \log \frac{n}{\varepsilon \delta})$ samples i.i.d. from $\mathcal{D}$ and if $(\sigma^*, p^*)$ minimizes $\mathsf{envy}_S$, then with probability at least $1 - \delta$, we have*

$$\mathsf{envy}_{\mathcal{D}}(\sigma^*, p^*) < \min_{(\sigma, p)} \mathsf{envy}_{\mathcal{D}}(\sigma, p) + \varepsilon.$$

Thus, if we draw sufficiently many samples, then with high probability the allocation minimizing expected envy on the sample will, up to $\varepsilon$, be minimizing with respect to $\mathcal{D}$.

We prove this result by discretizing the space of allocations. We then use a concentration inequality to show that w.h.p. the expected envy with respect to $\mathcal{D}$ is close to the expected envy with respect to the sample $S$. The proof of Theorem 5.1 appears in Appendix D.5.

Note that for this result we employed a direct approach. This technique and its discretization step would not have worked for the envy-free rate, because two very close rent divisions can in principle have very different EF rates. On the other hand, we expect that similar bounds for the minimum envy objective could be obtained by using extensions of VC dimension to real-valued functions (e.g., pseudo-dimension).

## 5.2 Computational Complexity

Again, our sample complexity result needs an algorithm that finds the best allocation for a given sample $S$. Like for EFrate, we can solve this problem using integer linear programming (see Appendix B). For the formal complexity analysis, consider the following decision problem:

---

EXPECTED ENVY MINIMIZATION
**Input:** List $S = (v^{(1)}, \ldots, v^{(m)})$, number $B$.
**Question**: Is there $(\sigma, p)$ with $\text{envy}_S(\sigma, p) \leqslant B$?

---

This problem is again NP-complete. The proof is in Appendix D.6, by reduction from CLIQUE.

**Theorem 5.2.** EXPECTED ENVY MINIMIZATION *is NP-complete, even for binary valuation profiles.*

Interestingly, this problem becomes easy once we fix a room assignment $\sigma$, because the best price vector can then be computed by linear programming (because the values of the integer variables in the ILP shown in Appendix B are decided by the fixed room assignment $\sigma$). Thus, the problem can be solved in time $n! \cdot \text{poly}(n, m)$, and hence is fixed-parameter tractable with respect to the number of agents $n$. This is good news: instances will often have a small number of agents, but we will want to consider as large a sample as feasible to ensure low maximum envy. In fact, as we will see momentarily, the NP-completeness of the problem is not an obstacle in real-world instances.

# 6 Experiments

We evaluated our rules on user data taken from Spliddit.[3] We studied distributions obtained by adding noise to valuations. We started by selecting 1,000 instances $v$ at random, to speed up computations. The same selection is used for each experiment. For each instance, we normalize the rent to 1, and normalize valuations to sum to 1. We considered three noise models, each parameterized by a choice of noise level $\varepsilon \in \{0, 0.01, \ldots, 0.09\}$.

$$v_{ir}^{(\ell)} \sim v_{ir} \cdot (1 + \text{Uniform}[-\varepsilon, +\varepsilon]) \tag{Uniform}$$

$$v_{ir}^{(\ell)} \sim v_{ir} \cdot (1 + \text{N}[0, \varepsilon]) \tag{Normal}$$

$$v_{ir}^{(\ell)} \sim v_{ir} \cdot (1 + r \cdot \text{N}[0, \varepsilon]) \tag{Biased Normal}$$

In each of these noise models, valuations are increased or decreased by a random fraction. Here, $N[\mu, \sigma]$ is a normal distribution with mean $\mu$ and standard deviation $\sigma$. For the biased normal noise model, we put rooms in an arbitrary fixed order and label them with integers $0, 1, \ldots, n-1$. Rooms with a higher index have more noise.

For each noise model and choice of $\varepsilon$, we produced a sample $S$ of size $m = 100$. We then computed allocations maximizing $\text{EFrate}_S$ and minimizing $\text{envy}_S$. We also computed the maximin and lexislack rules based on the input profile $v$. For each of the four allocations, we calculated their value of $\text{EFrate}_S$, and of $\text{envy}_S$. We then average over all 1,000 instances. The results are shown in Figure 2 for the Uniform noise model. Results for the other noise models and more details are given in Appendix A. As expected, on each of the two metrics, the rule optimizing it does best, but all three

---

[3] This dataset was kindly provided to us in anonymized form by the maintainer of Spliddit, Nisarg Shah.

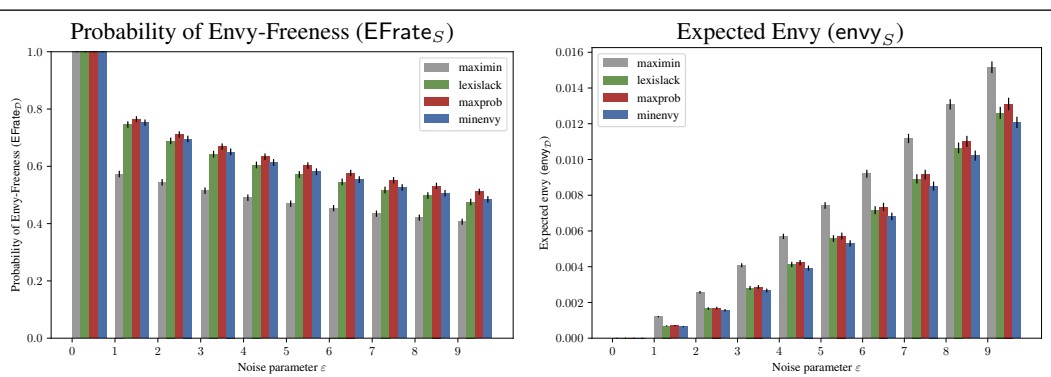

Figure 2: Results of experiments for the Uniform noise model.

rules aiming for robustness do similarly well. Spliddit's maximin rule does significantly worse on our metrics. Before the experiments, we expected that lexislack would do worse for the biased noise [model, but this does] not appear to be the case.

In the appendix, we also evaluate the sampling-based rules on a freshly drawn sample different from the sample used to optimize the rules (Appendix A.2) as well as on a sample drawn from a different probability distribution (Appendix A.3), to evaluate how sensitive these methods are to being optimized on a small sample and to knowing the 'right' noise distribution. In both cases, we find that the performance of the sampling-based methods worsens, while lexislack is still robust.

timing.multiplicative-normal.inst300.comps1

| Sample size | "min_envy" | "max_prob" |
|---|---|---|
| 1 | 0.002937862809437020 | 0.003075944309433300 |
| 25 | 0.171606457553183000 | 1.043525625551120 |
| 50 | 0.333838360182320 | 2.234962467836280 |
| 75 | 0.547860274302463 | 3.786158517583580 |
| 100 | 0.737325895343288 | 5.112192253226720 |
| 125 | 0.936284335770820 | 7.974426841660090 |

Figure 3 shows average computation time to compute allocations optimizing $\mathsf{EFrate}_S$ and $\mathsf{envy}_S$, using Gurobi 9.1.2 on four threads of an AMD Ryzen 2990WX (128 GB RAM) with the ILP formulations from Appendix B. The results were obtained for a random selection of 300 Spliddit instances with $n = 4$, with the Uniform noise model for $\varepsilon = 0.05$, and sample sizes $m$ varying from 1 to 125. Minimizing envy is much faster due to fewer integral variables. In Appendix A.4, we report some additional computation times as a function of the noise parameter $\varepsilon$.

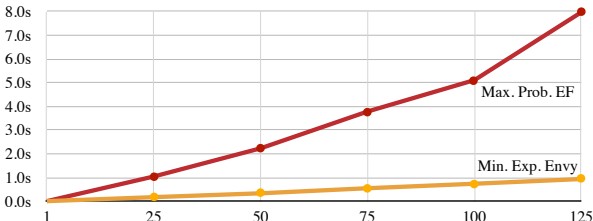

Figure 3: Computation time depending on sample size

# 7  Future Directions

Our approach should be applicable to many settings beyond rent division, such as homogeneous divisible goods, cake cutting, or even indivisible goods. For example, the lexislack rule can be adapted to these settings, and similar results as in our distribution-based approach might be achievable.

We have shown that the lexislack rule shares some key properties with the maximin rule, such as essential single-valuedness and polynomial-time computability. It would be interesting to axiomatically contrast the two solutions, for example with respect to strategic properties like manipulability.

For our distribution-based approach, we assumed that we have access to $\mathcal{D}$ only via sampling. Often we may know $\mathcal{D}$ more explicitly, for example if we are just adding noise to reported valuations. For such well-behaved $\mathcal{D}$, can we design direct algorithms for finding optimal allocations with respect to our two objectives, without needing samples?

## Acknowledgments and Disclosure of Funding

This work was partially supported by the National Science Foundation under grants IIS-2147187, CCF-2007080, IIS-2024287, and CCF-1733556; and by the Office of Naval Research under grant N00014-20-1-2488.

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
