# A Experimental Results

In this section, we will present more detailed results of the experiments described in Section 6 of the main body of the paper. We describe the results in four subsections:

- In Appendix A.1, we evaluate the rules by calculating their performance in terms of envy-rate and expected envy on the same sample ($m = 100$) that we used to optimize the two probabilistic rules (the *optimization sample*).

- In Appendix A.2, we evaluate the rules by calculating their performance on a fresh sample ($m = 1000$) that is different from the one used to compute the rules.

- In Appendix A.3, we evaluate our probabilistic rules trained on one specific noise model (normal noise with $\varepsilon = 0.05$) on the other noise models.

- In Appendix A.4 we briefly discuss computation times.

## A.1 Evaluation on the Optimization Sample

When evaluating the rules based on the optimization sample (or in other terminology, if we take the test set to be the same as the training set), then our optimizing rules are optimal by definition. Indeed, in each of the charts, we can see that the maxprob rule (maximizing the probability of envy-freeness) has the best performance out of all the rules with respect to the probability of envy-freeness; and analogously for the minenvy rule and the envy objective. When evaluating on the optimization sample, we see that the respective optimizing rule outperforms lexislack, but only by a modest amount.

In all the charts, error bars show standard errors, which are small due to the large number of instances.

### A.1.1 Uniform Noise

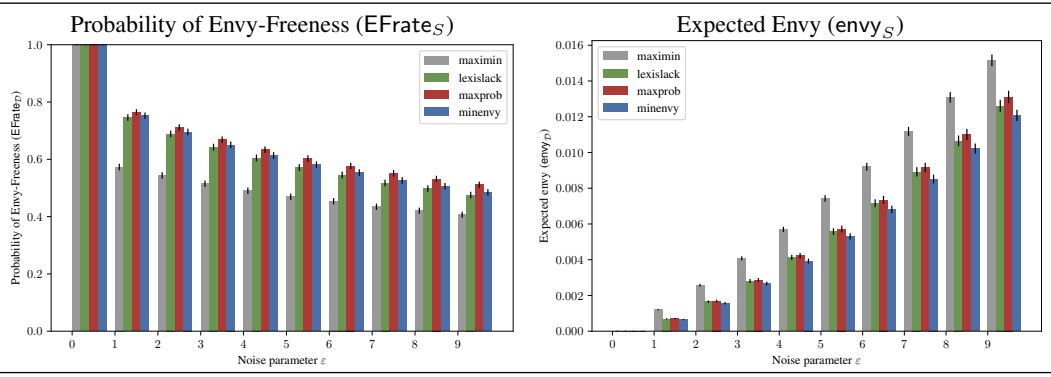

### A.1.2 Normal Noise

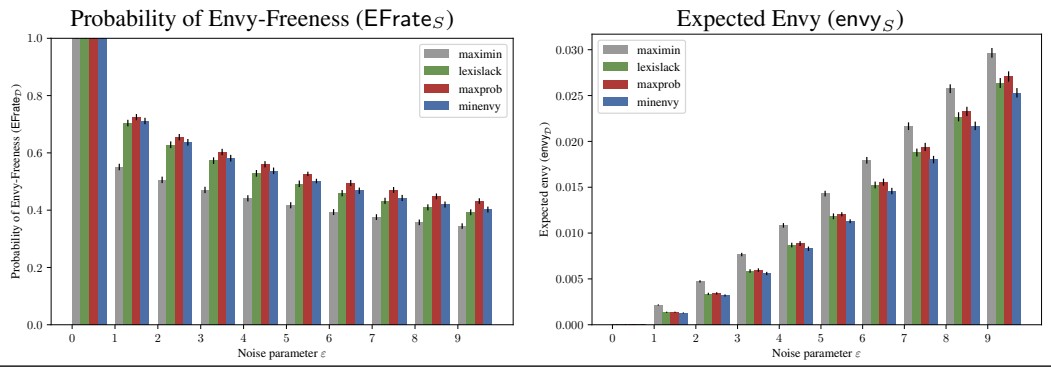

### A.1.3 Biased Normal Noise

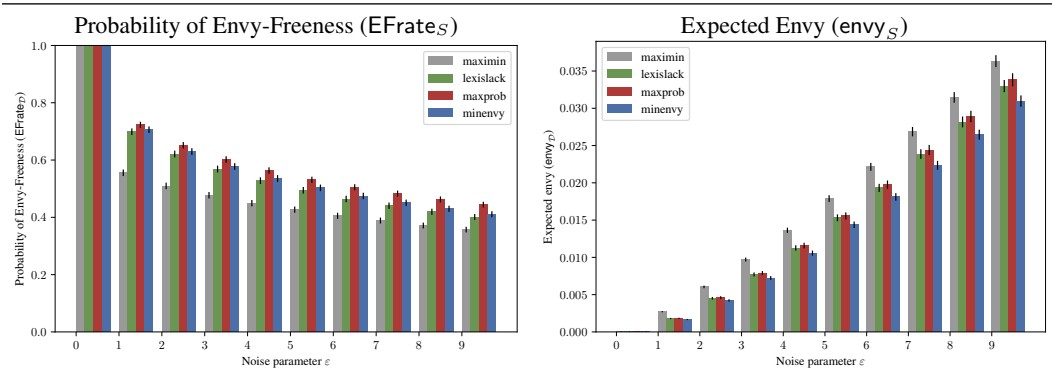

## A.2 Evaluation on a Fresh Sample

For these charts, we drew a fresh sample of size $m = 1000$, and calculated the values $\mathsf{EFrate}_S$ and $\mathsf{envy}_S$ with respect to this fresh sample. (Since evaluation is much cheaper than optimization, it is no problem to use a large sample size.) We see that the advantage of the optimizing rules over lexislack shrinks or disappears. The minenvy rule performs the same as lexislack for the uniform and normal noise models with respect to the envy objective, though it outperforms lexislack by an extremely small amount for the biased normal noise model. On the other hand, surprisingly, the maxprob rule does strictly worse than both the minenvy and lexislack rules, on the probability objective. This suggests that the sample size of $m = 100$ was too small to allow the maxprop rule to properly generalize to the underlying noise distribution.

### A.2.1 Uniform Noise

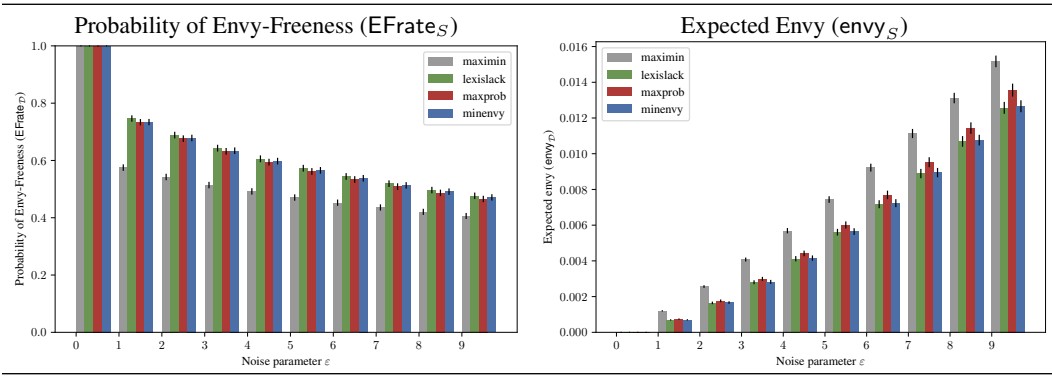

### A.2.2 Normal Noise

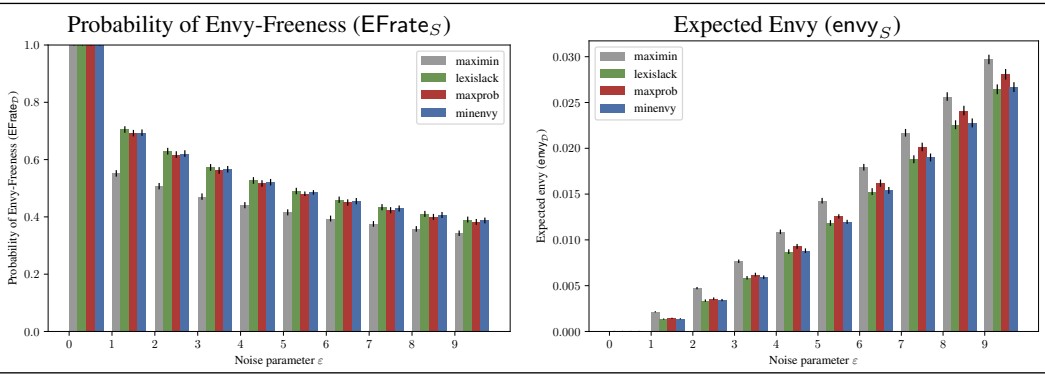

### A.2.3 Biased Normal Noise

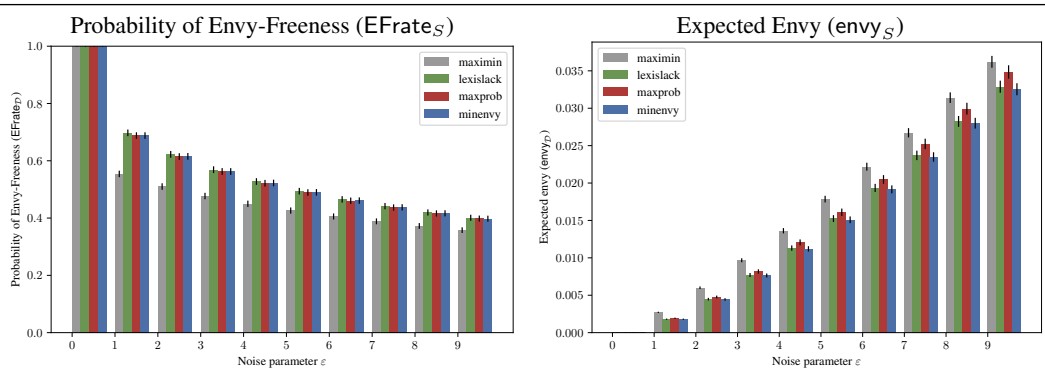

### A.3 Evaluation of Rules Trained on a Different Distribution

Like in the previous subsection, the following charts are with respect to a fresh sample of size $m = 1000$. However, in each chart, we now add two 'new' rules, namely the allocations selected by the maxprob and minenvy rules when optimized on samples drawn from the normal noise model with $\varepsilon = 0.05$. Thus, these charts allow us to gauge the performance of the distribution-based methods when they are optimized using the 'wrong' distribution.

For the probability objective and the uniform and normal noise models, not much performance is lost. However, the rules optimized for the normal noise models perform poorly for the biased noise models (compared to the rules optimized for that model, and also compared to lexislack). For the envy objective, we see notably bad performance for uniform noise.

### A.3.1 Uniform Noise

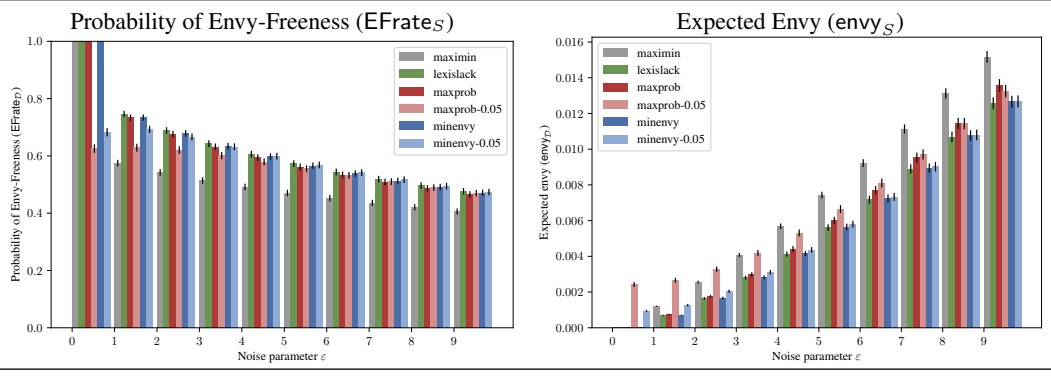

### A.3.2 Normal Noise

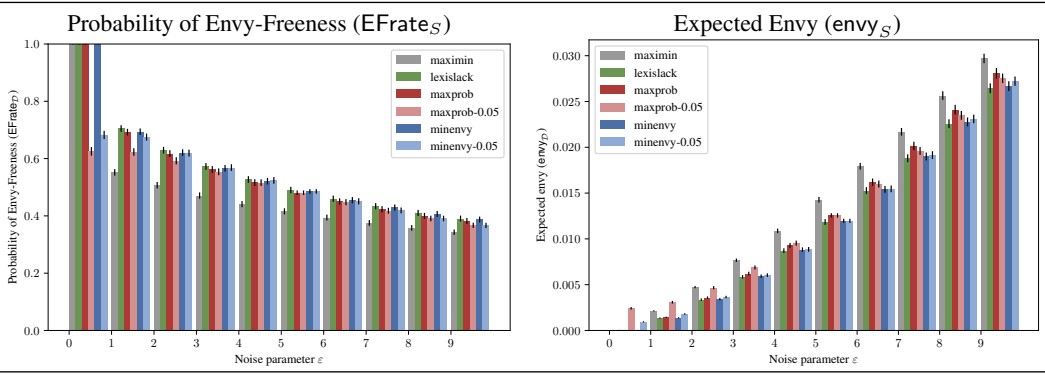

### A.3.3 Biased Normal Noise

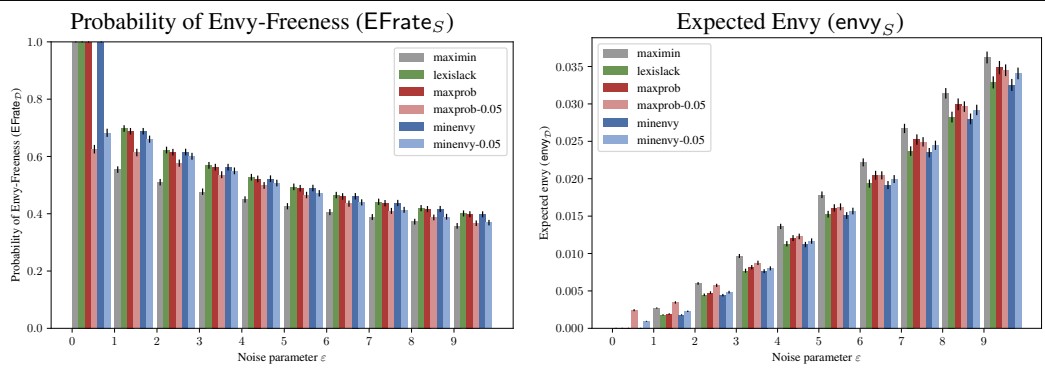

## A.4 Computation Time

In Figure 4, we show that the computation time of the two probabilistic rules when we vary the noise parameter $\varepsilon$ of the underlying distribution. The chart is based on computations involving 1000 instances from spliddit all with $n = 4$, and a sample size of $m = 100$. We can see that with zero noise ($\varepsilon = 0$), computation is extremely fast. This is because all the 100 samples are identical, so the ILP solver can eliminate most constraints as redundant. For positive noise ($\varepsilon \geqslant 0.1$), there is a very slight increase of computation time with increased noise.

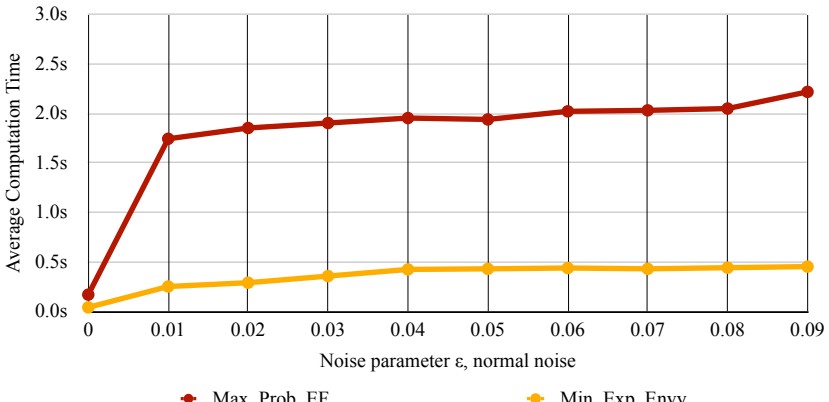

Figure 4: Computation time as a function of the noise parameter

# B  ILP Formulations

## B.1  Envy-Free Rate

Write $v_{\max} = \max_{i,\ell,r} v_{ir}^{(\ell)}$ (if we use normalized valuations, this value is at most 1). Note that an allocation $(\sigma, p)$ where $p_r < -v_{\max}$ for some $r \in R$ cannot be envy-free for any of the valuation profiles: If $r'$ is a room with $p_{r'} > 0$, then the person receiving $r'$ values $r'$ at most $v_{\max}$ more than room $r$, and hence by the price difference will envy the agent receiving room $r$. So in solving our maximization problem, we can restrict attention to price vectors with $p_r \geqslant -v_{\max}$ for all $r \in R$. Similarly we can assume $p_r \leqslant v_{\max}$.

Using such price vectors, note that the envy between any pair of players is at most $3v_{\max}$. Write $M = 3v_{\max}$.

$$
\begin{aligned}
\max \ & \sum_{\ell \in [m]} y_\ell \\
\text{s.t. } & \sum_{r \in R} x_{ir} = 1 && \text{for all } i \in N \\
& \sum_{i \in N} x_{ir} = 1 && \text{for all } r \in R \\
& v_{ir}^{(\ell)} - p_r \geqslant v_{ir'}^{(\ell)} - p_{r'} - M(1 - y_\ell) - M(1 - x_{ir}) && \text{for all } i \in N, r, r' \in R, \ell \in [m] \\
& \sum_{r \in R} p_r = 1 \\
& -v_{\max} \leqslant p_r \leqslant v_{\max} && \text{for all } r \in R \\
& x_{ir} \in \{0, 1\} && \text{for all } i \in N, r \in R \\
& y_\ell \in \{0, 1\} && \text{for all } \ell \in [m]
\end{aligned}
$$

## B.2  Minimize Expected Envy

As in the text, assume that valuations are normalized, and hence restrict attention to reasonable allocations with $-2 \leqslant p_r \leqslant 2$ for all $r$. Then envy is at most 5. Let $M = 5$.

$$
\begin{aligned}
\min \ & \sum_{\ell \in [m]} B_\ell \\
\text{s.t. } & \sum_{r \in R} x_{ir} = 1 && \text{for all } i \in N \\
& \sum_{i \in N} x_{ir} = 1 && \text{for all } r \in R \\
& (v_{ir'}^{(\ell)} - p_{r'}) - (v_{ir}^{(\ell)} - p_r) \leqslant B_\ell + M(1 - x_{ir}) && \text{for all } i \in N, r, r' \in R, \ell \in [m] \\
& \sum_{r \in R} p_r = 1 \\
& -2 \leqslant p_r \leqslant 2 && \text{for all } r \in R \\
& x_{ir} \in \{0, 1\} && \text{for all } i \in N, r \in R \\
& B_\ell \geqslant 0 && \text{for all } \ell \in [m]
\end{aligned}
$$

# C  Code

A brief overview of the code files included in the supplementary material.

rentdivisionmethods.py  An implementation of the rent division methods discussed in the paper: maximin (used on spliddit), lexislack, and the rules from the probabilistic context for maximizing probability of envy-freeness and for minimizing expected envy. Uses Gurobi as the underlying LP/ILP solver.

The other files are used for performing the experiments. Note that this depends on the data file from spliddit, which we do not have permission to share.

`rentdivisionmetrics.py` An implementation of metrics (envy rate, average envy, slack, social welfare): how good is a given allocation with respect to a valuation profile, or a sample/list of valuation profiles.

`neurips-experiments-step0-select-instances.py` Randomly orders the spliddit instances. We will compute the rules for the instances in that order.

`neurips-experiments-step1-compute-rules.py` Computes the outcome of the various rules on the spliddit instances. For the probabilistic rules, it goes through each model-noise combination, draws a sample, and computes the rule on those samples. The samples used and the rule outcomes are written to a pickle file.

`neurips-experiments-step2-analyze.py` The pickled results are read, and the various metrics are computed (with respect to the previously used sample or with respect to a fresh sample) and written to another pickle file.

`neurips-experiments-step3-plot.py` Plots are drawn using `matplotlib`.

We have included the outputs of the `step2-analyze` file, so that the `step3-plot` can be executed using a standard python3 installation with matplotlib and numpy installed.

## D Omitted Proofs

### D.1 Proof of Theorem 2.1

**Theorem.** *(a) For every optimal room assignment $\sigma$, there are prices $p$ so that $(\sigma, p)$ is envy-free. (b) If $(\sigma, p)$ is envy-free then $\sigma$ is optimal. (c) Let $\sigma_1$, $\sigma_2$ be optimal room assignments, and let $(\sigma_1, p)$ be an envy-free allocation. Then $(\sigma_2, p)$ is also an envy-free allocation, with all agents indifferent between the two: $v_{i\sigma_1(i)} - p_{\sigma_1(i)} = v_{i\sigma_2(i)} - p_{\sigma_2(i)}$ for all $i \in N$.*

*Proof.* (a) An optimal room assignment $\sigma$ forms a solution to the standard assignment problem [see, e.g., Wolsey, 1998, Section 4.3]. The dual of the assignment problem LP is

$$\min \sum_{i \in N} q_i + \sum_{r \in R} p_r \ \text{ s.t. } \ q_i + p_r \geqslant v_{ir} \ \text{ for } i \in N, r \in R.$$

Since $\sigma$ is an optimal room assignment, by complementary slackness there exists a solution $(q_i), (p_r)$ to the dual program where $q_i + p_{\sigma(i)} = v_{i\sigma(i)}$ for each $i \in N$. Thus

$$v_{i\sigma(i)} - p_{\sigma(i)} = q_i \geqslant v_{ir} - p_r \quad \text{for all } i \in N, r \in R,$$

using dual feasibility. Thus $(p_r)$ is an envy-free price vector, but we must ensure that $\sum_{r \in R} p_r = 1$, which we can do by adding a constant to each $p_r$. This preserves envy-freeness.

(b) Suppose $(\sigma, p)$ is envy-free and $\sigma'$ is any room assignment. Then $\sum_{i \in N} v_{i\sigma(i)} \geqslant \sum_{i \in N} (v_{i\sigma'(i)} - p_{\sigma'(i)} + p_{\sigma(i)}) = (\sum_{i \in N} v_{i\sigma'(i)}) - 1 + 1 = \sum_{i \in N} v_{i\sigma'(i)}$, where the inequality follows from envy-freeness. Thus $\sigma$ has at least the welfare of $\sigma'$. Since $\sigma'$ was arbitrary, $\sigma$ is an optimal assignment.

(c) We show $v_{i\sigma_1(i)} - p_{\sigma_1(i)} = v_{i\sigma_2(i)} - p_{\sigma_2(i)}$ for $i \in N$. From this, envy-freeness of $(\sigma_2, p)$ follows immediately. We have $v_{i\sigma_1(i)} - p_{\sigma_1(i)} \geqslant v_{i\sigma_2(i)} - p_{\sigma_2(i)}$ for all $i \in N$ since $(\sigma_1, p)$ is envy-free. Sum these inequalities to get

$$(\sum_{i \in N} v_{i\sigma_1(i)}) - 1 \geqslant (\sum_{i \in N} v_{i\sigma_2(i)}) - 1.$$

But the two sides of this inequality are equal, since both $\sigma_1$ and $\sigma_2$ are optimal. Hence each inequality is satisfied with equality, as required. □

### D.2 Proof of Theorem 3.2

**Theorem.** *The lexislack rule is essentially single-valued.*

*Proof.* For now, fix an optimal room assignment $\sigma$. We show that there is a unique price vector $p$ such that $(\sigma, p)$ is a lexislack solution. Because the leximin relation over vectors is strictly convex,

there is a unique vector $\Delta = (\Delta_{ir}(\sigma, p))_{i \in N, r \in R}$ maximizing the leximin objective, since if there were two different ones, a convex combination of the two would be strictly better. But $\Delta$ uniquely specifies a price vector: $\Delta$ specifies the differences $p_r - p_{r'}$ between any pair of prices, and with $\sum_r p_r = 1$ this gives a unique price vector.

Next, we show that if $\sigma_1$ and $\sigma_2$ are optimal room assignments, and $(\sigma_1, p)$ is an envy-free allocation, then $\Delta(\sigma_1, p) = \Delta(\sigma_2, p)$. By Theorem 2.1(c), $(\sigma_2, p)$ is an allocation where every agent obtains the same utility as under $(\sigma_1, p)$. Let $i \in N$. If $\sigma_1(i) = \sigma_2(i)$, then clearly the values $\Delta_{ir}$ are the same in both allocations. If $r_1 = \sigma_1(i) \neq \sigma_2(i) = r_2$, then equal utility under both allocations implies $v_{ir_1} - p_{r_1} = v_{ir_2} - p_{r_2}$, and hence $\Delta_{ir_2}(\sigma_1, p) = 0$ and $\Delta_{ir_1}(\sigma_2, p) = 0$. By definition, also $\Delta_{ir_1}(\sigma_1, p) = \Delta_{ir_2}(\sigma_2, p) = 0$, so the values of $\Delta_{ir_1}$ and $\Delta_{ir_2}$ agree on both allocations. For $r \in R \setminus \{r_1, r_2\}$, we have that the value of $\Delta_{ir}$ agrees on both allocations by the equal utility property. Hence $\Delta(\sigma_1, p) = \Delta(\sigma_2, p)$.

Thus, any vector $\Delta \geqslant 0$ achievable on one optimal room assignment can be achieved on any other optimal room assignment, with the same utility vector. This holds in particular for the lexislack vector $\Delta$. We have seen that for any fixed room assignment, there is a unique lexislack utility vector. Hence the lexislack utility vector is unique. $\square$

## D.3 Proof of Theorem 4.5

**Theorem.** EF-RATE MAXIMIZATION *is NP-complete, even for binary valuation profiles (where* $v_{ir} \in \{0, 1\}$ *for all $i$ and $r$).*

*Proof.* Membership in NP is clear. We give a reduction from CLIQUE. Let $G = (V, E)$ be a graph with $n$ vertices and $m$ edges and let $k$ be the target clique size.

We make each vertex an agent, $N = V$. The set of rooms is $R = \{r_1, \ldots, r_k, d_1, \ldots, d_{n-k}\}$ consisting of $k$ *slot rooms* and of $n - k$ *dummy rooms*. Writing $E = \{e_1, \ldots, e_m\}$, we construct $m$ valuation profiles, one per edge. For $\ell \in [m]$, write $e_\ell = \{u, v\}$; the valuation profile $v^{(\ell)}$ is defined by

$$v_{i,r}^{(\ell)} = \begin{cases} 1 & \text{if } i \in \{u, v\} \text{ and } r \in \{r_1, \ldots, r_k\}, \\ 0 & \text{otherwise.} \end{cases}$$

Thus, in the $\ell$th valuation profile, the two agents corresponding to the endpoints of the $\ell$th edge want to be in a slot room. All other agents do not care. Finally, set $B = \binom{k}{2}$.

We prove that $G$ has a $k$-clique iff there is an allocation that is envy-free in at least $B$ of the valuation profiles.

($\Leftarrow$): Suppose $(\sigma, p)$ is envy-free for $B$ profiles. Let $C \subseteq V$ be the set of $k$ agents/vertices that are assigned to slot rooms under $\sigma$; write $C = \{i_1, \ldots, i_k\}$. Let $\ell_1, \ldots, \ell_B$ be the collection of indices corresponding to valuation profiles under which the allocation is envy-free. We claim that $C$ is a clique, by showing that $e_{\ell_t} \subseteq C$ for each $t \in [B]$. This suffices since a set of $k$ vertices with $\binom{k}{2}$ edges is a clique.

Let $t \in [B]$. Since $(\sigma, p)$ is envy-free under $v^{(\ell_t)}$, by Theorem 2.1(b), $\sigma$ is optimal under $v^{(\ell_t)}$. This implies that $\sigma$ has welfare 2, which happens only if both endpoints of edge $e_{\ell_t}$ get a slot room. So by definition of $C$, $e_{\ell_t} \subseteq C$, as desired.

($\Rightarrow$): Suppose there is a clique $C \subseteq V$ of size $k$ in $G$; write $C = \{i_1, \ldots, i_k\}$. Make a room assignment $\sigma$ in which we assign agent $i_s$ to slot room $r_s$, for each $s \in [k]$. The remaining agents can be assigned arbitrarily to dummy rooms. We set $p_r = \frac{1}{n}$ for each $r \in R$.

Write $e_{\ell_1}, \ldots, e_{\ell_B}$ for the set of edges within $C$; there are exactly $B$ of them since $C$ is a clique. Let $t \in [B]$, and write $e_{\ell_t} = \{i_a, i_b\}$. We show that $(\sigma, p)$ is envy-free under $v^{(\ell_t)}$. All agents except $i_a$ and $i_b$ are indifferent between all rooms, and since all rents are the same, they are not envious. Agents $i_a$ and $i_b$ both receive a room that they most prefer, and since all rents are the same, they are not envious. $\square$

## D.4 Proof of Theorem 4.6

**Theorem.** EF-RATE MAXIMIZATION WITH FIXED ASSIGNMENT *is NP-complete.*

*Proof.* Membership in NP is clear. We give a reduction from FEEDBACK ARC SET, which can be stated as follows.

---

**Input:** Digraph $D = (V, E)$, number $B$.
**Question:** Is there an ordering $(x_1, \ldots, x_n)$ of $V$ such that at least $B$ arcs from $E$ point from left to right? (An arc $(x_k \to x_s) \in E$ points from left to right if $k < s$.)

---

Consider an instance of this problem: Let $D = (V, E)$ be a digraph and let $B$ be a number. We construct a rent division instance where $V$ is both the set of agents and of rooms. Let $\sigma(x) = x$ be the identity room assignment.

Let $\varepsilon = 2/(n(n+1))$, chosen so that $\varepsilon + 2\varepsilon + \cdots + n\varepsilon = 1$.

Label the arc set $E = \{a_1, \ldots, a_m\}$. We define one valuation profile for each arc $a_\ell = (x \to y)$, with

$$v_{xy}^{(\ell)} = 1 + \varepsilon, \qquad v_{zz}^{(\ell)} = 1 \text{ for all } z \in V,$$

and valuation 0 for all unspecified combinations.

We now prove that there is an ordering $(x_1, \ldots, x_n)$ of $V$ with at least $B$ arcs pointing from left to right if and only if there exists a price vector $p$ that makes the identity room assignment $\sigma$ envy-free in at least $B$ of the valuation profiles.

($\Rightarrow$): Let $(x_1, \ldots, x_n)$ of $V$ be an ordering such that (wlog) the arcs $a_1, \ldots, a_B$ point from left to right.

Consider the price vector $p = (\varepsilon, 2\varepsilon, \ldots, n\varepsilon)$, so room $x_i$ has rent $i \cdot \varepsilon$. This is a valid price vector because it sums up to 1 by choice of $\varepsilon$. We claim that this price vector is envy-free for valuation profiles $v^{(1)}, \ldots, v^{(B)}$. Let $\ell \in [B]$. Since $a_\ell$ points from left to right, we have $a_\ell = (x_k \to x_s)$ for some $k < s$. First, note that any agent $x_i \neq x_k$ does not envy another agent because $x_i$ values her assigned room $x_i$ at utility 1 higher than other rooms, which avoids envy because room prices differ by less than 1. By the same argument, agent $x_k$ never envies any agent except perhaps $x_s$. Finally, we check that agent $x_k$ does not envy agent $x_s$: Note that the rent of room $x_s$ is $s \cdot \varepsilon$, which is at least $\varepsilon$ higher than the rent $k \cdot \varepsilon$ of room $x_k$. Since $x_k$ values room $x_s$ only $\varepsilon$ more than her assigned room $x_k$, she does not envy agent $x_s$. Thus $p$ is envy-free for valuation profile $v^{(\ell)}$, as required.

($\Leftarrow$): Suppose there is a price vector $p$ that is envy-free for valuation profiles $v^{(1)}, \ldots, v^{(B)}$ (relabeled for convenience). Label the vertices $x_1, \ldots, x_n$ in order of increasing price, i.e., such that $p_{x_1} \leqslant p_{x_2} \leqslant \cdots \leqslant p_{x_n}$ with ties broken arbitrarily. Let $\ell \in [B]$ and consider arc $a_\ell = (x_k \to x_s)$. We show that $a_\ell$ points from left to right, i.e., $k < s$. As $p$ is envy-free for agent $x_k$ under $v^{(\ell)}$, we have

$$v_{x_k, x_s}^{(\ell)} - p_{x_s} \leqslant v_{x_k, x_k}^{(\ell)} - p_{x_k} \iff 1 + \varepsilon - p_s \leqslant 1 - p_k$$
$$\iff p_{x_s} \geqslant p_{x_k} + \varepsilon.$$

In particular $p_{x_k} < p_{x_s}$. By our choice of ordering, it follows that $k < s$, as required. $\qquad\square$

## D.5 Proof of Theorem 5.1

**Theorem.** *Let $\varepsilon, \delta > 0$, and let $\mathcal{D}$ be a distribution. If we draw $m = O(\frac{n}{\varepsilon^2} \log \frac{n}{\varepsilon\delta})$ samples i.i.d. from $\mathcal{D}$ and if $(\sigma^*, p^*)$ minimizes $\mathsf{envy}_S$, then with probability at least $1 - \delta$, we have*

$$\mathsf{envy}_{\mathcal{D}}(\sigma^*, p^*) < \min_{(\sigma, p)} \mathsf{envy}_{\mathcal{D}}(\sigma, p) + \varepsilon.$$

We start by proving a few technical lemmas. First, define $\Lambda$ to be the set all allocations $(\sigma, p)$ with $-2 \leqslant p_r \leqslant 2$ for all $r \in R$. We call such allocations *reasonable*. Our first lemma shows that we may restrict attention to reasonable allocations only: in particular, if an allocation minimizes $\mathsf{envy}_{\mathcal{D}}$ then it must be reasonable.

**Lemma D.1.** *Let $\sigma$ be a room assignment and $v$ a profile.*

(a) *If $p$ is a price vector with $|p_r - p'_r| > 2$ for some $r, r' \in R$, then $\mathsf{envy}_v(\sigma, p) > 1$.*

(b) *If $p = (\frac{1}{n}, \ldots, \frac{1}{n})$, then $\mathsf{envy}_v(\sigma, p) \leqslant 1$.*

(c) *If $(\sigma, p)$ is reasonable, then $\mathsf{envy}_v(\sigma, p) \leqslant 5$.*

*Proof.* Note that since valuations are assumed to sum to 1, we have $v_{ir} - v_{ir'} \leqslant 1$ for all $r, r' \in R$.

(a) Say $p_r > p_{r'} + 2$ and $\sigma(i) = r$. Since $v_{ir} \leqslant v_{ir'} + 1$ as just noted, we have $v_{ir} - p_r < v_{ir'} - p_{r'} - 1$.

(b) If prices for all rooms are equal, then $\mathsf{envy}_v(\sigma, p) = \max\{0, \max_{i,j}(v_{i\sigma(j)} - v_{i\sigma(i)})\} \leqslant \max\{0, 1\} = 1$.

(c) $|v_{i\sigma(j)} - v_{i\sigma(i)}| + |p_{\sigma(j)} - p_{\sigma(i)}| \leqslant 1 + 4 = 5.$ $\qquad\square$

From now on, we assume all allocations to be reasonable.

Our second lemma says that if two allocations have similar price vectors, then they have similar expected envy.

**Lemma D.2.** *Let $p, p' \in \mathbb{R}^n$ be such that $|p_r - p'_r| \leqslant t$ for all $r \in R$. Then for any sample $S$ and distribution $\mathcal{D}$,*

$$|\mathsf{envy}_{\mathcal{D}}(\sigma, p) - \mathsf{envy}_{\mathcal{D}}(\sigma, p')| \leqslant 2t,$$
$$|\mathsf{envy}_S(\sigma, p) - \mathsf{envy}_S(\sigma, p')| \leqslant 2t.$$

*Proof.* First, we claim that for any valuation profile $v$,

$$|\mathsf{envy}_v(\sigma, p) - \mathsf{envy}_v(\sigma, p')| \leqslant 2t.$$

This holds since the value of $(v_{i\sigma(j)} - p_{\sigma(j)}) - (v_{i\sigma(i)} - p_{\sigma(i)})$ changes by at most $\pm 2t$ if we move from $p$ to $p'$, and thus the same holds after taking the maximum.

Now let $\mathcal{D}$ be a distribution. By linearity of expectation, and since $|\mathbb{E}[X]| \leqslant \mathbb{E}[|X|]$ by Jensen's inequality,

$$|\mathsf{envy}_{\mathcal{D}}(\sigma, p) - \mathsf{envy}_{\mathcal{D}}(\sigma, p')|$$
$$= |\mathbb{E}_{v \sim \mathcal{D}}[\mathsf{envy}_v(\sigma, p) - \mathsf{envy}_v(\sigma, p')]|$$
$$\leqslant \mathbb{E}_{v \sim \mathcal{D}}\left[|\mathsf{envy}_v(\sigma, p) - \mathsf{envy}_v(\sigma, p')|\right] \leqslant 2t,$$

where the last inequality follows by our claim. This proves the first statement. The second statement follows from the first by taking $\mathcal{D}$ to be the uniform distribution over $S$. $\qquad\square$

We also need a standard concentration inequality.

**Lemma D.3** (Hoeffding's inequality). *Let $X_1, \ldots, X_m$ be i.i.d. random variables with $0 \leqslant X_k \leqslant c$ and $\mathbb{E}[X_k] = \mu$ for all $k \in [m]$. Then for all $\varepsilon > 0$,*

$$\Pr\left[|\mu - \tfrac{1}{m}\sum_{k=1}^m X_i| \geqslant \varepsilon\right] \leqslant 2\exp(-2m\varepsilon^2/c^2).$$

We are now ready to prove our main result of this section.

*Proof of Theorem 5.1.* Let $t = 1/\lceil 12/\varepsilon \rceil$. Let $\Lambda^t \subseteq \Lambda$ be the set of all *discretized allocations* $(\sigma, p)$ where $p_r$ is an integer multiple of $t$. Note that for any $(\sigma, p) \in \Lambda$, there is a discretized allocation $(\sigma, p') \in \Lambda^t$ with $|p_r - p'_r| \leqslant t$ for all $r$ (call such allocations *t-close*), obtained by rounding the values $p_r$ up or down to ensure that $\sum_r p'_r = 1$.

Let $S$ be a random sample from $\mathcal{D}$ of size $m$, where

$$m = \tfrac{200}{\varepsilon^2} \ln\left[\left(\tfrac{60}{\varepsilon}\right)^n \tfrac{2n!}{\delta}\right] = O\left(\tfrac{n}{\varepsilon^2} \ln\left(\tfrac{n}{\varepsilon\delta}\right)\right).$$

Now write:

- $\mathrm{OPT}_{\mathcal{D}}$ for an allocation $(\sigma, p)$ minimizing $\mathsf{envy}_{\mathcal{D}}$,
- $\mathrm{OPT}_S$ for an allocation minimizing $\mathsf{envy}_S$ (which depends on the random choice of $S$),
- $\overline{\mathrm{OPT}}_{\mathcal{D}} \in \Lambda^t$ for a discretized allocation $t$-close to $\mathrm{OPT}_{\mathcal{D}}$,
- $\overline{\mathrm{OPT}}_S \in \Lambda^t$ for a discretized allocation $t$-close to $\mathrm{OPT}_S$.

Let $(\sigma, p) \in \Lambda$. For $k \in [m]$, let $X_k$ be the random variable taking the value $\mathsf{envy}_v(\sigma, p)$, where $v$ is the $k$th sample in $S$. By reasonableness and Lemma D.1, $0 \leqslant X_k \leqslant 5$. Then Hoeffding's inequality implies that

$$\Pr\left[|\mathsf{envy}_S(\sigma, p) - \mathsf{envy}_{\mathcal{D}}(\sigma, p)| \geqslant \tfrac{\varepsilon}{4}\right] \leqslant 2\exp(-\tfrac{2}{25}m(\tfrac{\varepsilon}{4})^2).$$

Let $E$ be the event that $|\mathsf{envy}_S(\sigma, p) - \mathsf{envy}_{\mathcal{D}}(\sigma, p)| < \varepsilon/4$ holds for all discretized allocations $(\sigma, p) \in \Lambda^t$ simultaneously. By Hoeffding's inequality and a union bound over all $|\Lambda^t| \leqslant (\tfrac{4}{t})^n n!$ discretized allocations, we get

$$\Pr[E] \geqslant 1 - (\tfrac{4}{t})^n n! \, 2\exp(-\tfrac{2}{25}m(\tfrac{\varepsilon}{4})^2) \geqslant 1 - \delta.$$

where the second inequality holds by choice of $m$.

Suppose that the event $E$ attains. In this case we have:

$$
\begin{aligned}
\mathsf{envy}_{\mathcal{D}}&(\text{OPT}_S) - \mathsf{envy}_{\mathcal{D}}(\text{OPT}_{\mathcal{D}}) & \\
&= (\mathsf{envy}_{\mathcal{D}}(\text{OPT}_S) - \mathsf{envy}_{\mathcal{D}}(\overline{\text{OPT}}_S)) & \text{(Lemma D.2)}\\
&\quad + (\mathsf{envy}_{\mathcal{D}}(\overline{\text{OPT}}_S) - \mathsf{envy}_S(\overline{\text{OPT}}_S)) & (E \text{ attains})\\
&\quad + (\mathsf{envy}_S(\overline{\text{OPT}}_S) - \mathsf{envy}_S(\text{OPT}_S)) & \text{(Lemma D.2)}\\
&\quad + (\mathsf{envy}_S(\text{OPT}_S) - \mathsf{envy}_S(\overline{\text{OPT}}_{\mathcal{D}})) & \text{(optimality)}\\
&\quad + (\mathsf{envy}_S(\overline{\text{OPT}}_{\mathcal{D}}) - \mathsf{envy}_{\mathcal{D}}(\overline{\text{OPT}}_{\mathcal{D}})) & (E \text{ attains})\\
&\quad + (\mathsf{envy}_{\mathcal{D}}(\overline{\text{OPT}}_{\mathcal{D}}) - \mathsf{envy}_{\mathcal{D}}(\text{OPT}_{\mathcal{D}})) & \text{(Lemma D.2)}\\
&< 2t + \varepsilon/4 + 2t + 0 + \varepsilon/4 + 2t & \\
&= 6/\lceil 12/\varepsilon \rceil + \varepsilon/2 \leqslant \varepsilon. &
\end{aligned}
$$

The references on the right indicate what we have used to bound the respective term to obtain the strict inequality. "Optimality" refers to the fact that $\text{OPT}_S$ minimizes $\mathsf{envy}_S$. Because event $E$ implies the above inequality, we see that

$$\Pr[\mathsf{envy}_{\mathcal{D}}(\text{OPT}_S) - \mathsf{envy}_{\mathcal{D}}(\text{OPT}_{\mathcal{D}}) < \varepsilon] \geqslant \Pr[E] \geqslant 1 - \delta.$$

This proves the result. $\qquad\square$

## D.6 Proof of Theorem 5.2

**Theorem.** EXPECTED ENVY MINIMIZATION *is NP-complete, even for binary valuation profiles.*

*Proof.* Membership in NP is clear. Reduction from CLIQUE.

Let $G = (V, E)$ be a graph with $n$ vertices and $m$ edges and target clique size $k$. Set the target envy amount to be $B = m - \binom{k}{2}$. Let $M$ be a large integer, $M > (B+1)^2$. Write $\varepsilon = (B+1)/M$. Our choices of these numbers imply the following estimates which we will need later:

- $M\varepsilon > B$, since $M\varepsilon = B + 1 > B$.

- $\varepsilon(B+1) < 1$, since $\varepsilon(B+1) = (B+1)^2/M < 1$.

The set of agents is $V$. The set of rooms is $R = \{o_1, \ldots, o_k, d_1, \ldots, d_{n-k}\}$ consisting of $k$ *slot rooms* and $n - k$ dummy rooms. Write $E = \{e_1, \ldots, e_m\}$. We construct a sample $S$ of $m + M$ valuation profiles. For $j \in [m]$, write $e_j = \{u, v\}$; then valuation profile $v^{(j)}$ is defined by

$$v_{i,r}^{(j)} = \begin{cases} 1 & \text{if } i \in \{u, v\} \text{ and } r \in \{o_1, \ldots, o_k\}, \\ 0 & \text{otherwise.} \end{cases}$$

For $j = m+1, \ldots, m+M$, let $v^{(j)}$ be a *uniform profile*:

$$v_{i,r}^{(j)} = 0 \quad \text{for all } i \in V \text{ and } r \in R.$$

($\Rightarrow$): Suppose there is a clique $C \subseteq V$ of size $k$ in $G$; write $C = \{i_1, \ldots, i_k\}$. We construct an allocation $(\sigma, p)$ that will be envy-free for $B$ profiles. In the room assignment, we will assign agent $i_r$ to slot room $o_r$, for $r \in [k]$. The remaining agents can be assigned arbitrarily to dummy rooms. We'll say that each room costs the same rent so $p_r = \tfrac{1}{n}$ for all $r \in R$.

- For each of the $M$ uniform profiles, $\text{envy}_v(\sigma, p) = 0$.

- For a profile $v$ corresponding to an edge $e_j = \{i_a, i_b\}$ with $i_a, i_b \in C$ (i.e. contained in the clique), $\text{envy}_v = 0$.

- For one of the $m - \binom{k}{2}$ profiles $v$ corresponding to edges not contained in a clique, we have $\text{envy}_v = 1$.

Summing these up, we have $\text{envy}_S(\sigma, p) = m - \binom{k}{2} = B$.

($\Leftarrow$): Suppose there is an allocation $(\sigma, p)$ with $\text{envy}_S(\sigma, p) \leqslant B$. Let $C \subseteq V$ be the set of $k$ agents/vertices assigned to slot rooms under $\sigma$; write $C = \{i_1, \ldots, i_k\}$.

First we show that the rents $p = (p_1, \ldots, p_n)$ of the rooms are close to uniform, in the sense that $|p_r - p_{r'}| \leqslant \varepsilon$ for all $r, r' \in R$. Assume for a contradiction that there are $r, r' \in R$ with $p_r > p_{r'} + \varepsilon$. Then in each uniform profile, the agent assigned to room $r$ envies the agent assigned to room $r'$ by at least $\varepsilon$, and hence the max envy in a uniform profile is at least $\varepsilon$. Since we have introduced $M$ uniform profiles, it follows that $\text{envy}_S(\sigma, p) \geqslant M\varepsilon > B$, a contradiction.

Now we show that $C$ is a clique. Suppose not. Then there are at least $m - \binom{k}{2} + 1 = B + 1$ edges that are not completely contained in $C$. For each profile corresponding to such an edge, the agent corresponding to the endpoint not in $C$ envies other agents who are assigned a slot room by at least $1 - \varepsilon$. Hence $\text{envy}_S(\sigma, p) \geqslant (B+1)(1-\varepsilon) = B + 1 - \varepsilon(B+1) > B$, because $\varepsilon(B+1) < 1$. This is a contradiction. $\qquad\square$