# OpenReview forum: "Robust Rent Division"
_NeurIPS.cc/2022/Conference — NeurIPS 2022 Accept_

### Official Review · Reviewer_5Apq · 2022-07-06

**Rating:** 8
**Confidence:** 4
**Soundness:** 4 excellent
**Presentation:** 3 good
**Contribution:** 4 excellent

**Summary:**

This paper proposes three robust approaches for the envy-free rent division problem: the lexislack approach that looks for allocations that remain envy-free for a range of valuations around the input one, an approach that maximizes the probability of envy-freeness, and one that takes the magnitude of envy into consideration by minimizing it. The authors provide time complexities for obtaining solutions for all these approaches, as well as the probabilistic bounds that guarantee the quality of the solutions of the latter two approaches. The robustness and the computation times of the proposed approaches are demonstrated on user data collected from Spliddit.

**Questions:**


Below are some detailed suggestions and questions for the paper.

$\bullet$ Line $116$, Page $3$ (Section $2$): what does "weakly prefers" mean?

$\bullet$ Lines $129$ to $130$, Page $3$ (Section $2$): "we can restrict... of the form $(\sigma,p)$" is a little bit confusing to me: isn't an allocation always of the form $(\sigma,p)$?

$\bullet$ Formula $(1)$, Page $4$ (Section $3$): it seems to me that it should be $(v_{i\sigma(i)}-v'_{i\sigma(i)})+(v'_{i\sigma(j)}-v_{i\sigma(j)})\leq s$.

$\bullet$ Line $146$, Page $4$ (Section $3$): at the end of the proof, it is better to recall that the conclusion holds since $i,j\in N$ are arbitrary.

$\bullet$ Theorem $4.1$, Page $5$ (Section $4$): I would suggest providing a brief description of the spirit of Theorem $4.1$ in the main text.

$\bullet$ Lines $199$ to $200$, Page $5$ (Section $4$): I would suggest providing references for the sentence "In learning theory...need not hold."

$\bullet$ Line $200$, Page $5$ (Section $4$): I guess the "realizability assumption" corresponds to the setting "the ground truth $\tau$ need not be a member of $\mathcal{H}$". Please elaborate these two more clearly.

$\bullet$ Line $207$, Page $5$ (Section $4$): "...with ${\rm VCdim}(\mathcal{H})=d$" should be "...with ${\rm VCdim}(\mathcal{H})=d$ over $X$"?

$\bullet$ Line $258$, Page $7$ (Section $4$): could the authors explain why "Then with probability $1$...assignment $\sigma^{(\ell)}$" is a direct result of the continuous distribution assumption?

$\bullet$ Line~$261$, Page $7$ (Section $4$): the correspondence between the i$th$ call ($i\in[m]$) and the $\sigma$ (that is optimal for $v^{(i)}$) should be stated clearly.

$\bullet$ Lines $296$ to $298$, Page $8$ (Section $5$): it might be better to move "On the other hand...pseudo-dimension)." to the last section.

$\bullet$ Line $307$, Page $8$ (Section $5$): could the authors provide more detail about this linear program?

$\bullet$ Line $307$, Page $8$ (Section $5$): I think "this means that" is incorrect here: "computed by linear programming" does not "means" that the problem can be solved with certain time complexity.

$\bullet$ Lines $317$ to $318$, Page $8$ (Section $6$): what do the superscripts $(\ell)$'s mean?

$\bullet$ Lines $324$ to $326$, Page $9$ (Sectio $6$): the authors should point out clearly that "For each of... over all $1000$ instances" is the testing phase.

$\bullet$ Lines~$329$ to $330$, Page~$9$ (Section~$6$): the sentence "Before the...to be the case." is unnecessary.

$\bullet$ Figure~$2$, Page~$9$ (Section~$6$): I suggest the authors report the computation time with more values of $n$.

**Limitations:**

Line $108$, Page $3$ (Section $2$): "...each agent $i\in N$ and each room $r\in R$" should be "..each $(i,r)\in N\times R$"?

Lines $137$ to $138$, Page $4$ (Section $3$): in the formulation of "${\rm slack}$", "$\Delta_{ir}$" should be "$\Delta_{ir}(\sigma, p)$"?

Line $164$, Page $4$ (Section $3$): "$n^2$ values in $\Delta(\sigma,p)$" should be "$\{\Delta(\sigma,p)\}_{(\sigma,p)\in N^2}$".

Line~$231$, Page $6$ (Section $4$): "pair $i,j\in N$" better to say "pair $(i,j)\in N \times N$"?

There are several other typos or grammatical errors. I would suggest proofreading the paper carefully.

**Strengths And Weaknesses:**

Strengths:
The models proposed in this paper are very practical. My favourite part is the establishment of Theorem $4.1$, wherein the authors intelligently parallel a standard result in PAC learning to their framework and derive the probabilistic bound for their solution.

Weaknesses:
Some places can be better elaborated and several typos should be corrected.

---

> ### Author Response · Authors · 2022-07-29
> **Author Response**
>
> Thank you for your careful reading. We will fix the typos and other minor issues that you have identified.
>
> * Line 116: "weakly prefers x to y" means "the utility of x is at least as large as the utility of y".
> * Line 129-130: Yes, all allocations can be written as $(\sigma, p)$; the point of this sentence is that we may fix $\sigma$ to one particular room assignment.
> * Line 258 ("with probability 1 there is a unique optimal room assignment"): This is because there are finitely many possible room assignments and the social welfare of each one is drawn from a continuous distribution, so, with probability 1, no two assignments will have the same social welfare. (This is not a formal proof but hopefully the intuition is clear.)
> * Line 307 (more details about the LP): this is the LP obtained by taking the ILP from the appendix and explicitly setting the correct values of the binary variables of that ILP. We will briefly mention this.
> * Line 307 ("this means that"): Linear programs (without integer variables) can be solved in polynomial time (e.g. by the ellipsoid method), so our claim here is correct.

---

### Official Review · Reviewer_NxEV · 2022-07-06

**Rating:** 6
**Confidence:** 4
**Soundness:** 4 excellent
**Presentation:** 4 excellent
**Contribution:** 2 fair

**Summary:**

The paper studies the problem of robust envy-free rent division.  In the conventional problem variant, given n individuals and their evaluations for n rooms, the goal is to find a rule to allocate the rooms and fairly split the rent such that everyone is satisfied with the room and the split payment (i.e., allocation is envy-free).  Prior literature assumes that the individuals' valuations are accurate. This paper proposes robust allocation rules such that the allocation remains envy-free (or maximizes the probability of being envy-free, or minimizes expected envy) even if the reported evaluations deviate from the true ones.

**Questions:**

My main question is whether the authors gave any thought to the implications of their approaches to robust fair rent devision on incentives to misreport values (i.e., are these stronger in this context, and how do the different models of "robustness" impact incentives).

**Strengths And Weaknesses:**

Strengths:

1. The paper is very well-written, and the organization of the results make it easy to follow. The paper studies several notions of robustness against different uncertainty around the true evaluations.  The l1 notion seems quite natural, and yields a polynomial-time algorithm.  It seems that l_inf robustness would do similarly, although the paper doesn't state that explicitly (just that Prop 3.1 essentially goes through for l_inf robustness).  Distributional variations are interesting as well; although they yield NP-hard optimization problems (but poly sample complexity), the associated ILPs seem fast enough in practice.

2. There is a particularly interesting sample complexity result for the distributional variant of this problem in which the authors derive VC dimension of the allocation "hypothesis space" (which is polynomial).  This nicely combines with relatively standard arguments to get polynomial sample complexity.

Weakness:
1. I found the l_1 and l_inf models of robustness considerably more convincing than the distributional models, for a few reasons.  First, l_inf seems to address the main limitation of l_1 that the authors highlight in the introduction.  Second, l_p models capture a nice intuition related to incentive compatibility: that while people may misreport valuations, they will typically stick relatively close to true valuations (although admittedly here the Gaussian model also makes good sense).  This is not to say that the distributional models are bad; it's simply that they seem somewhat forced here, in the sense that added value seems low, and increased complexity therefore too weakly motivated.

2. There isn't any discussion of incentive compatibility.  This is natural here, since it conflicts with envy-freeness (if I recall correctly), but it would still be useful to understand how the notions of robustness impact manipulability, at least one compared to the other.

3. It would also be useful to include computational results for Lexislack, perhaps as a part of Fig 2 (if it makes sense visually), or separately.

---

> ### Author Response · Authors · 2022-07-29
> **Author Response**
>
> Thank you for your feedback.
>
> > There isn't any discussion of incentive compatibility. This is natural here, since it conflicts with envy-freeness (if I recall correctly), but it would still be useful to understand how the notions of robustness impact manipulability, at least one compared to the other. [...] My main question is whether the authors gave any thought to the implications of their approaches to robust fair rent devision on incentives to misreport values (i.e., are these stronger in this context, and how do the different models of "robustness" impact incentives).
>
> You recall correctly that envy-freeness and strategyproofness are known to be incompatible [Alkan et al. 1991, Appendix]. We did not think about how egalitarianism and robustness differ in their implications on manipulability, in part because, in our motivating applications like Spliddit, strategic behavior is not a major issue. The reason is that, while the website goes to great lengths to explain fairness guarantees to laypeople, the vast majority of users will not even bother to learn the details of the technical solutions that are used (which would require reading scientific articles). Without this knowledge, users are unlikely to be able to reason about how strategic manipulation would affect the outcome.
>
> In response to your remark, we did some brief simulations to check whether there is a clear difference in manipulability between maximin and lexislack, but the results are inconclusive for now. (That is, these initial results are sensitive to the exact choice of manipulation metric and to the choice of sample.) So — to the extent that strategic behavior is a concern — we would leave this question for future work.
>
> > It would also be useful to include computational results for Lexislack, perhaps as a part of Fig 2 (if it makes sense visually), or separately.
>
> We did not include lexislack in Figure 2 because the line would be indistinguishable from the x-axis. Spliddit instances are not big enough for lexislack computation time to become a concern, but we will think about running computation time experiments with synthetic data (which we have not used in the other experiments).
>
> > There seems to be a small typo: should the right-hand EFrate_D be EFrate_S?
>
> Thanks, do you remember which line you were referring to here? If it is line 189.5, then EFrate_D is correct.

---

### Official Review · Reviewer_Vr2D · 2022-07-11

**Rating:** 8
**Confidence:** 4
**Ethics Flag:** Yes
**Soundness:** 3 good
**Presentation:** 4 excellent
**Contribution:** 4 excellent

**Summary:**

This paper focuses on the problem of fair rent division: how does one split a total cost among agents in a way that respects the individual agents' valuations and returns an allocation and a cost split that is envy free? This paper focuses on the case where there is uncertainty in the valuations reported by the agents. The paper therefore introduces what they call the lexi-slack solution which remains envy free with as large a radius as possible from the agent valuation reports. They also look at the case where we get samples from a valuation distribution and we want to maximize the probability of selecting an EF allocation with a small number of samples.

**Questions:**

* A major major issue I have with the paper as it currently exists is around the use of data. No discussion of IRB process, informed consent etc. is given for the data from Spliddit. To my knowledge this data is not open source and it is not clear that the use of this data was cleared with users from the website. I hope the authors addressed this in a responsible way, otherwise I do not feel the results can or should be published, ethically. I have flagged this paper for this reason and hope the authors have these issues handled and just didn't include details in this writeup.

* See question above about practical applications of the sampling results.

* See question about other heuristics for refining the solution space of maximin that could be used/evaluated in a more fair manner than the "better" claim made in the paper.

**Ethics Review Area:**

["Privacy and Security (e.g., consent)"]

**Limitations:**

One question / limitation that isn't discussed but might be interesting for future work is any notion of the mechanism being adversarial. The writing currently points to the fact that maximin can cause envy for small changes in valuations but what if the center is choosing these in a way to benefit one agent or another. This isn't a hard limitation but the discussion is a bit taught in part 7 so a bit more would be nice.

Building on this, there is a discussion of the limitations of the L_1 distance metric, were other metrics used? Why L_1 and not others given that even the authors say that L_1 can over optimize for one agent... a bit more discussion would be nice.

**Strengths And Weaknesses:**

++ The paper is extremely well written, it is easy to follow, and all problems and notation are clear. The discursive style is very nice and overall it was a very fun and interesting paper to read.

++ The proofs and results seem correct to me and are written very clearly. While I did not check the proofs in detail I was able to follow most of the arguments and they seemed reasonable.

++ I really enjoyed the sample / learning portion of the paper and found that interesting. One question/limitation I would have liked to see discussed is how this sampling would happen in the real world? Would one agent report their valuations multiple times??? I don't see how this would be practically applicable.

-- I find the claim: "We end with some experiments on data taken from Spliddit. They suggest that our three new rules
83 significantly outperform the Spliddit maximin rule on robustness metrics." a little disingenuous. Of course the method that returns an EF allocation that has as a secondary constraint these other robustness metrics is better, this is not surprising. What would greatly strengthen the claim is comparing to some other heuristically optimized robustness metric. While this is really not that necessary since lexislack is basically this (since it doesn't minimize expected envy which is NP-hard) but this claim could be written in a more even handed way and/or evaluated against other poly-time heuristics that get stacked on the maximin rule.

-- None of the experiments contain error bars or confidence intervals for especially Figure 1. So it's basically impossible to tell what the difference are and whether or not they are significantly different. The empirical claims would be much improved with some error bars or empirical confidence intervals.

---

> ### Author Response · Authors · 2022-07-29
> **Author Response**
>
> Thank you for your feedback.
>
> > A major major issue I have with the paper as it currently exists is around the use of data. No discussion of IRB process, informed consent etc. is given for the data from Spliddit. To my knowledge this data is not open source and it is not clear that the use of this data was cleared with users from the website. I hope the authors addressed this in a responsible way, otherwise I do not feel the results can or should be published, ethically. I have flagged this paper for this reason and hope the authors have these issues handled and just didn't include details in this writeup.
>
> Your point is well taken. We should have mentioned (and will mention) that the dataset was obtained from the developers of Spliddit, and, crucially, that the dataset contains no personally identifiable information (i.e., the data we interacted with are fully anonymized).
>
> Regarding the IRB process, we can confidently say that our simulations are exempt from IRB approval at our institution. Indeed, the guidelines state that IRB approval is needed only when the study involves "a living individual about whom an investigator conducting research obtains (1) information or biospecimens through intervention or interaction with the individual and uses, studies, or analyzes the information or biospecimens or (2) obtains, uses, studies, analyzes, or generates identifiable private information or identifiable biospecimens." Point (1) doesn't apply because there was no interaction, and (2) doesn't apply because no identifiable information was obtained or used.
>
> It's worth noting that, while the dataset isn't publicly available, it has been shared responsibly with experts in fair division; in fact, it has been used in a number of previous papers, starting from the influential 2016 paper "the unreasonable fairness of maximum Nash welfare." For the purposes of evaluating our own paper, we believe the ethical question should be whether there is a realistic possibility that the publication of the empirical results (summary statistics aggregated over numerous instances) would violate the privacy of Spliddit users or harm them in some other way; in our view, the answer is quite clearly negative.
>
> > I really enjoyed the sample / learning portion of the paper and found that interesting. One question/limitation I would have liked to see discussed is how this sampling would happen in the real world? Would one agent report their valuations multiple times??? I don't see how this would be practically applicable.
>
> Please see our discussion in lines 66-73 and 77-81, where we explain that our proposal is to use *synthetic* samples: we impute a  distribution such as Gaussian noise centered at the reported valuations, and then use a computer to sample repeatedly from this distribution. Thus, in this scheme, agents report valuations only once.
>
> > None of the experiments contain error bars or confidence intervals for especially Figure 1. So it's basically impossible to tell what the difference are and whether or not they are significantly different. The empirical claims would be much improved with some error bars or empirical confidence intervals.
>
> We noticed this omission after submission. There are error bars in the figures in the supplementary material. We will update the figures in the main body to use the same style.
>
> > Of course the method that returns an EF allocation that has as a secondary constraint these other robustness metrics is better, this is not surprising.
>
> While we agree it is not surprising that our methods outperform maximin on robustness metrics, this is not obvious and so needed to be established. For example, in principle it could have been that on real instances, there are very few envy-free outcomes that don't differ much in terms of robustness. It is worth noting that we evaluate the methods on metrics that are not explicitly optimized by the method, for example when lexislack turns out to perform well with respect to the distributional models.
>
> > See question about other heuristics for refining the solution space of maximin that could be used/evaluated in a more fair manner than the "better" claim made in the paper.
>
> Note that the maximin method selects only one outcome ("essentially single-valued", Alkan et al. 1991) so it is not possible to refine this solution for robustness. It may be possible to define methods that trade off between egalitarian welfare and robustness, which may be interesting in future work.
>
> > Building on this, there is a discussion of the limitations of the L_1 distance metric, were other metrics used? Why L_1 and not others given that even the authors say that L_1 can over optimize for one agent... a bit more discussion would be nice.
>
> Thanks, we will add more discussion. We briefly mention L_infinity in line 147.

---

> > ### Comment · Reviewer_Vr2D · 2022-08-06
> > **Two Issues**
> >
> > To be clear, I really like this paper and think the work is fantastic.
> >
> > >>I really enjoyed the sample / learning portion of the paper and found that interesting. One question/limitation I would have liked to see discussed is how this sampling would happen in the real world? Would one agent report their valuations multiple times??? I don't see how this would be practically applicable.
> >
> > >Please see our discussion in lines 66-73 and 77-81, where we explain that our proposal is to use synthetic samples: we impute a distribution such as Gaussian noise centered at the reported valuations, and then use a computer to sample repeatedly from this distribution. Thus, in this scheme, agents report valuations only once.
> >
> > -- Ah okay, I think I kinda get it, but is there justification for Gaussian here? I guess I was looking for something about using either between or in-subjects sampling instead of just blurring and re-sampling.
> >
> > ## RE: IRB.
> >
> > It's above my pay grade to make this call but yes of course the simulations (synthetic data) do not need IRB review. However, "we can confidently say that our simulations are exempt from IRB approval at our institution" is not acceptable or okay. I've done many many studies with data that I collected -- you should have an exemption from the IRB from your institution and that number should be provided as part of this user study since this data is not public. I agree that this study is most likely exempt, I'd honestly be surprised if it wasn't, but again, there is a process for this that you did not follow. The AC/SPC/Ethics review whomever can make the call here but if you're using human subjects data that you collected or is not public then researcher promises are not enough, it needs to be reviewed by the IRB.

---

> > > ### Author Response · Authors · 2022-08-07
> > > **Two Responses**
> > >
> > > **Regarding sampling:** There is no particular justification for Gaussians here beyond it being a natural default way to add noise. Our experiments suggest that the choice of noise model (e.g. changing the variance slightly, or going for uniform instead) does not change the results very much (Appendix A.3). Needless to say, our theoretical results are not specific to Gaussians, rather they apply to any distribution. Therefore, in cases where a different distribution emerges in an application (even one induced by an interactive sampling scheme), our approach will provide the same guarantees.
> > >
> > > **Regarding IRB:** We greatly appreciate your attention to this issue, and we have ourselves always sought and received IRB approval for any experiments that needed approval; but, to the best of our understanding, the study in this paper does not require approval.
> > >
> > > Perhaps it’s useful to elaborate on the kind of data we used as part of our simulations. Spliddit is a website that is used by everyday people to share rent (as well as other applications). Users input their utilities and the website recommends a solution using a particular algorithm (“maximin”). A Spliddit developer who is not an author provided us with an anonymized copy of the input utilities going back to 2014. We then computed what room allocations would have been recommended to the users by different algorithms. Note that, in the course of this study, we had no contact with users, and all sampling is synthetic.
> > >
> > > Because there was no interaction with users and we do not have access to personally identifiable information, we did not do “human subjects research” as defined by federal regulations in the US. The instructions by our IRB (“decision tree”) specify that neither a review nor an exemption are needed for such data analysis (whether a dataset is public or not does not feature as part of the decision tree). Other institutions may handle this differently, but we did follow the process applicable to us, to the best of our understanding.

---

### Official Review · Reviewer_4a6w · 2022-07-21

**Rating:** 8
**Confidence:** 4
**Soundness:** 4 excellent
**Presentation:** 3 good
**Contribution:** 3 good

**Summary:**

This paper provides new algorithmic and complexity results for the problem of computing envy-free solutions to the fair division problem. Here, n agents must each be matched to one of n rooms given their valuations for the rooms and split the total rent for the n rooms among them, and the agents have quasilinear utilities, i.e., their utility for receiving a room is its value minus their payment which is their share of the rent. Previous work proves that an envy-free solution to the rent division problem with quasilinear utilities always exists and can be computed in polynomial time using a linear program to compute the prices that result in envy-freeness. However, existing approaches to compute such envy-free allocations are not robust to perturbations to agent's utilities (such as the Spliddit algorithm which is focused on maximizing the utility of the agent with lowest utility).

The main conceptual contribution of this paper is a new objective: maximizing robustness to perturbations in agent utilities under the envy-freeness constraint. Two such objectives are considered under a Gaussian noise model, i.e., when agent's true valuations are assumed to be "drawn" from a distribution obtained by adding Gaussian noise to their reported valuations:
- Maximizing the probability that the allocation is envy-free w.r.t. this distribution
- Minimizing the expected envy, i.e., the sum total of envy experienced between every possible pair of agents, w.r.t. the distribution

Given a set of samples from this distribution, there are polynomial time algorithms that compute the optimal solution w.r.t. either of these objectives and the set of samples.

The main technical contributions are:
- Sample complexity results for the number of samples needed to provide a guarantee that the solution computed by a polynomial time algorithm is at least within some tolerance parameter of the optimal solution with high probability for either of the above objectives.
- Computational hardness results for computing the optimal solutions w.r.t. the distribution of valuation profiles

**Questions:**

None

**Limitations:**

I believe the authors have adequately addressed the limitations of this work and proposed meaningful avenues for future work.

**Strengths And Weaknesses:**

Strengths:
- [Relevance] The paper uses several tools from learning theory to arrive at significant technical results.
- [Novelty] The proposed objectives of maximizing robustness are novel to the best of my knowledge.
- [Significance] The paper makes several important and interesting conceptual and technical contributions which I believe will be of interest to the large computation social choice community. The future directions pointed out are well chosen and I agree that they are interesting questions for future work (e.g. how to elicit uncertain valuations from users on platforms like Spliddit and their implications on computing robust solutions)
- [Writing] The paper is well written and organized and I agree with the authors' choice of which proofs are included in the main paper or left to the appendix. The discussions of the motivation and related work are sufficient given the space constraints.
- [Soundness] I was able to verify the main results and they seem sound. However, I was unable to check all of the results (e.g. the computational hardness results) thoroughly due to a lack of time.

Weaknesses:
- No significant weaknesses that must be addressed in a conference publication.
- It would have been interesting to see at least experimental results with more noise models. However, this is a minor nitpick given the space constraints.

---

> ### Author Response · Authors · 2022-07-29
> **Author Response**
>
> Thank you for your feedback.
>
> > It would have been interesting to see at least experimental results with more noise models. However, this is a minor nitpick given the space constraints.
>
> We agree. If (like last year) the camera-ready version is allowed an additional page, we will prioritize moving experiments from the appendix into the main body.

---

### Review · Ethics_Reviewer_u6JT · 2022-08-05

**Recommendation:**

The authors should say more about their data set or provide a citation that describes it.  In particular, the authors should not just say that it was de-identified, but say what's left so that reviewers can judge whether enough de-identification happened.

Now being aware of the lack of notice, ideally, the authors would not use the old data for future research and would start using only data collected after the website is updated with a privacy policy mentioning research use.  I recommend suggesting this change to the maintainers of Spliddit.  However, these recommendations must be balanced against the availability of new data and the ability to replicate results.


**Ethical Issues:**

Yes

**Ethics Review:**

A review flagged this submission for an ethics review under the issue of "Privacy and Security (e.g., consent)".  The review asks whether the data about users of Spliddit was ethically used, such as collected with consent and under IRB oversight.  This is a reasonable ethical concern to raise.

The authors comment that the data is exempt from institutional review since they didn't create the data by interacting with the users and the data had no identifiable information.  I would have more confidence in this claim if I knew what data the authors got.  For example, I noticed that Spliddit collects street addresses to fairly split cab fares.  Did the authors' receive that data?

Regardless of whether the authors complied with IRB requirements (or the lack thereof), the data use might be unethical for other reasons.  I could not find a privacy policy or any notice on the website for Spliddit that the data might be used for research.  This raises two ethical issues: a lack of notice and a lack of consent.

About the lack of notice, the website is pretty clearly an academic project, which at least hints at the possibility of research use of the data.  Still it would be better to be explicit about such.

About the lack of consent, without notice, it's hard to argue that the users gave even implied consent.  However, consent doesn't seem strictly needed (even ethically, again, I'm not currently discussing IRB requirements).  To put this into context, HIPAA allows the use of de-identified medical data for research without consent.  However, a HIPAA Notice of Privacy Practices should at least tell patents about this use, which brings me back to the lack of notice.

I don't think the lack of notice is so egregious as to make publishing this submission unethical.  However, now that the authors are aware of it, they should try to mitigate the situation.  I admit that this compromise opinion is a judgement call and that others may find the lack of notice so egregious as to make publication unethical or so trivial as to not be worth even mentioning it.

---

### Review · Ethics_Reviewer_hTZi · 2022-08-13

**Recommendation:**

Should the authors write an ethical considerations section where they highlight these data aspects, I see no issues moving forward with this.

**Ethics Review:**

I don’t really see any ethical issues beyond the use of potentially sensitive data from spliddit, which the authors ought to write a quick section on how the data was acquired, handled, any IRB applications, etc.

---

### Meta-Review · Area_Chair_E8S3 · 2022-08-26

**Recommendation:** Accept
**Confidence:** Certain

**Metareview:**

Reviewers are all positive and excited about the paper: interesting and natural model, novel and robust mechanism with theoretical guarantees, nice sample complexity analysis, experiments on real-world data.

**Award:**

No

---

### Decision · Program_Chairs · 2022-09-14

Accept